

# Sinomacrops bondei, a new anurognathid pterosaur from the Jurassic of China and comments on the group

Xuefang Wei[1,2,3], Rodrigo Vargas Pêgas[4], Caizhi Shen[5], Yanfang Guo[5], Waisum Ma[6], Deyu Sun[7] and Xuanyu Zhou[8,9]

[1] Key Laboratory of Stratigraphy and Palaeontology, Ministry of Natural Resource, Institute of Geology, Chinese Academy of Geological Sciences, Beijing, China
[2] China University of Geosciences, Beijing, China
[3] Centre of Cores and Samples of Nature Resources, China Geological Survey, Beijing, China
[4] Federal University of ABC, São Bernardo, Brazil
[5] Dalian Natural History Museum, Dalian, Liaoning, China
[6] School of Geography, Earth and Environmental Sciences, University of Birmingham, Birmingham, UK
[7] Jinzhou Paleontology Museum, Jinzhou, Liaoning, China
[8] Department of Natural History Sciences, Hokkaido University, Sapporo, Japan
[9] Beipiao Pterosaur Museum of China, Beipiao, Liaoning, China

## ABSTRACT

Anurognathids are an elusive group of diminutive, potentially arboreal pterosaurs. Even though their monophyly has been well-supported, their intrarelationships have been obscure, and their phylogenetic placement even more. In the present work, we present a new genus and species from the Middle-Late Jurassic Tiaojishan Formation, the third nominal anurognathid species from the Jurassic of China. The new species provides new information concerning morphological diversity for the group. Furthermore, we provide a new phylogenetic analysis incorporating into a single data set characters from diverging phylogenetic proposals. Our results place them as the sister-group of Darwinoptera + Pterodactyloidea, as basal members of the Monofenestrata.

## INTRODUCTION

Pterosaurs, a group of archosauromorph reptiles of disputed placement (see *Renesto & Binelli, 2006*; *Hone & Benton, 2007*; *Bennett, 2013*; *Ezcurra et al., 2020*), were the first vertebrates known to develop active flight, with a fossil record stretching from the Late Triassic to the K/Pg boundary (*Wellnhofer, 1991*; *Andres, Clark & Xu, 2014*; *Dalla Vecchia, 2014*). The Anurognathidae are a very peculiar pterosaur group still poorly understood and rather obscure, characterized by a unique morphology and involved in a complex history of uncertainty about their phylogenetic affinities (*Hone, 2020*). Spanning from the Middle Jurassic (Callovian) to the Early Cretaceous (Aptian), anurognathids are small-sized (up to 900 mm in wingspan) and exhibit short skulls with a diminutive preorbital region, huge orbits and rounded jaws that are wider than long (*Bennett, 2007*;

Corresponding author
Xuanyu Zhou, xyzhou@elms.hokudai.ac.jp

*Hone, 2020*). Due to their short wings with low aspect ratios and their peg-like teeth, these small pterosaurs have been interpreted as aerial insectivores (*Bennett, 2007*; *Witton, 2008*, *2013*; *Ősi, 2011*; *Habib, 2011*; *Hone, 2020*), of possible arboreal habits (*Ji & Ji, 1998*; *Bennett, 2007*; *Witton, 2013*; *Lü et al., 2018*; *Hone, 2020*).

The Anurognathidae have been defined as a node-based group, as the least inclusive clade containing *Anurognathus ammoni* and *Batrachognathus volans* (*Kellner, 2003*; *Unwin, 2003*). Recently, it has been redefined as a branch-based group, englobing all species closer to *Anurognathus* than to *Dimorphodon*, *Pterodactylus* or *Scaphognathus* (*Hone, 2020*). So far, this group comprises six nominal species, and is known by 12 specimens from Germany, Kazakhstan, Mongolia, China and North Korea (with a putative 13th one from the USA). The first described one was *Anurognathus ammoni*, coming from the Tithonian Solnhofen limestones of Bavaria (*Döderlein, 1923*) and being represented by two specimens (*Bennett, 2007*). It was not until the second specimen was described that several aspects of its morphology were clarified, such as the broad wings, the short preorbital region and extensive orbit, the jugal overlying the maxilla, the vertical (or slightly anteriorly inclined) quadrate, the reduced palatal elements, and the short tail lacking filiform processes of the zygapophyses and haemapophyses, convergent with pterodactyloids (*Bennett, 2007*).

The second nominal species was *Batrachognathus volans*, described from an incomplete skeleton including a partial skull from the Oxfordian-Kimmeridgian Karabastau Formation of Kazakhstan (*Riabinin, 1948*). A second specimen of *Batrachognathus volans* (*Unwin, Lü & Bakhurina, 2000*), still awaiting a full description, possesses a tail that bears well-developed rod-like processes of the haemapophyses and zygapophyses, and is longer than that of any other anurognathid (*Costa et al., 2013*). With this discovery, *Batrachognathus volans* became the first known anurognathid to exhibit a long tail with developed rod-like processes as typical of most non-pterodactyloid pterosaurs (see *Costa et al., 2013*).

The third anurognathid to be described was *Dendrorhynchoides curvidentatus*, the first recovered from a Cretaceous deposit, the early Aptian Jianshangou beds of the Yixian Formation (*Ji & Ji, 1998*). Originally thought of as Barremian, these beds are now viewed as early Aptian in age (see *Chang et al., 2009*).

*Jeholopterus ningchengensis*, based on an almost complete skeleton with extensive soft tissue preservation coming from the Daohugou beds near Daohugou (Ningcheng County, Inner Mongolia), was later described as another Cretaceous anurognathid (*Wang et al., 2002*), on the basis of the now outdated view of the Daouhugou beds as part of the Yixian Formation (Barremian-Aptian). Subsequently, these beds were reinterpreted as part of the Middle-Late Jurassic Tiaojishan Formation. Presently, these rocks have been once more reinterpreted, and are now considered to belong to the Haifanggou/Jiulongshan Formation (*Huang, 2015*, *2016*). The locality that has yielded *Jeholopterus ningchengensis* has been dated as Callovian-Oxfordian (*Liu, Liu & Yang, 2006*; *Gao & Shubin, 2012*). A second specimen from the same locality has been regarded as most likely conspecific with *Jeholopterus ningchengensis*, though a detailed description and a formal taxonomic assessment have not been provided yet (*Ji & Yuan, 2002*; *Witton, 2013*; *Yang et al., 2019*).

Later, a second species for the genus *Dendrorhynchoides*, named *D. mutoudengensis*, was erected based on an almost complete skeleton from the Mutoudeng locality, Tiaojishan Formation (*Lü & Hone, 2012*). Recently, a new genus has been erected to accommodate this species: *Luopterus*, named after the late Prof. Junchang Lü (*Hone, 2020*). Moreover, a second Cretaceous anurognathid was also named recently, *Vesperopterylus lamadongensis*, known from an almost complete holotype from the late Aptian Jiufotang Formation (*Lü et al., 2018*).

Indeterminate specimens include IVPP V16728, which stands out as the second specimen with a long tail and developed rod-like processes, similar to *Batrachognathus volans* (see *Costa et al., 2013*) and unlike all remaining anurognathids. NJU–57003 is another long-tailed specimen from the Mutoudeng locality (Daohugou Beds, Tiaojishan Formation), only preliminarily described (*Yang et al., 2019*). A relatively complete specimen from the Early Cretaceous of North Korea also awaits description (*Gao et al., 2009*), as well as a fragmentary specimen comprised of wing elements from the Middle Jurassic (Aalenian/Bajocian) Bakhar deposits of Central Mongolia (*Bakhurina & Unwin, 1995*). Finally, the poorly-known *Mesadactylus ornithosphyos*, based on the holotype BYU 2024 (a synsacrum) from the Kimmeridgian-Tithonian Morrison Formation of the USA (*Jensen & Padian, 1989*), is a potential anurognathid (see *Bennett, 2007*).

Pterosaur phylogeny is intricated with controversies, but no other group compares to the Anurognathidae when it comes to uncertainty concerning its placement (*Young, 1964*; *Unwin, 1992*, *1995*, *2003*; *Viscardi et al., 1999*; *Kellner, 2003*; *Andres, Clark & Xing, 2010*; *Dalla Vecchia, 2014*, *2019*; *Hone, 2020*). Five cladistic hypotheses based on computed analyses have been presented for the Anurognathidae, wherein they are viewed as: the basalmost pterosaur group (*Kellner, 2003*); the sister-group of the Novialoidea (*Unwin, 2003*); the sister-group of the Breviquartossa (*Dalla Vecchia, 2019*); scaphognathids, whereby these are the sister-group of the Monofenestrata (*Vidovic & Martill, 2017*); or the sister-group of the Pterodactyloidea (*Andres, Clark & Xing, 2010*; *Andres, Clark & Xu, 2014*). And even though the monophyly of the Anurognathidae has been strongly corroborated (*Kellner, 2003*; *Unwin, 2003*; *Bennett, 2007*; *Andres, Clark & Xing, 2010*; *Dalla Vecchia, 2019*), its intrarelationships have been poorly explored (*Hone, 2020*).

This work presents a new fossil coming from the Mutoudeng locality, JZMP-2107500095, representing a new genus and species of long-tailed anurognathid. Despite being crushed to the point of obliterating many details, the specimen is rather complete and provides new information for the group, including the first record of an anurognathid skull exposed in mostly lateral view. In other specimens, the skull is either exposed in mostly internal view, as in the holotype of *Anurognathus ammoni* (*Döderlein, 1923*; *Wellnhofer, 1975*; *Bennett, 2007*), or dorsoventrally crushed, as in all other specimens that preserve a skull (*Riabinin, 1948*; *Ji & Ji, 1998*; *Bennett, 2007*; *Gao et al., 2009*; *Lü & Hone, 2012*; *Lü et al., 2018*).

We further review the phylogenetic relationships of the group (both intra and inter), presenting an analysis including all proposed species and a resulting in a new hypothesis for the placement of the group as basal monofenestratans.

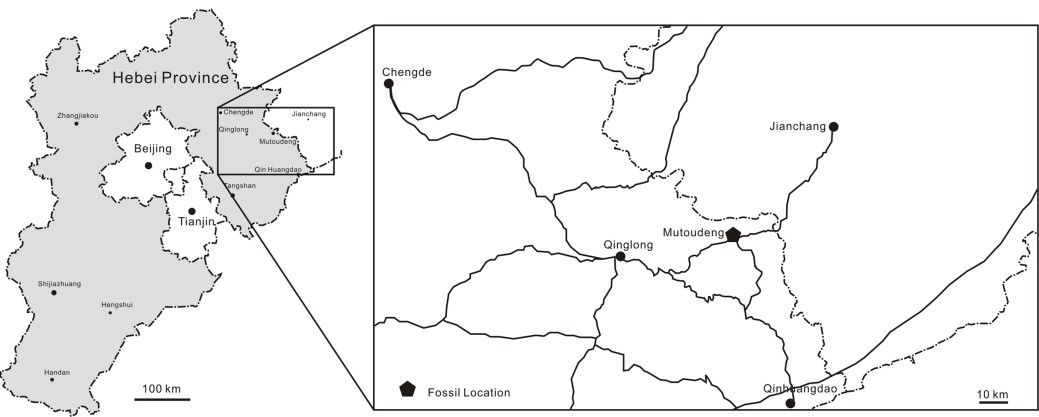

**Figure 1 Fossil provenance.** Maps indicating Hebei Province (China). JPM-2012-001 comes from the Mutoudeng locality.

## GEOLOGICAL SETTING

The Tiaojishan Formation takes its name from the Tiaojishan Mountain (Mentougou District, Beijing), and was named by *Ye (1920)*. This and the Haifanggou/Jiulongshan Formation have yielded the famous Yanliao Biota in western Liaoning and adjacent regions (*Huang, 2015*, *2016*). This biota is well known for the beautiful preservation and abundancy of insects and vertebrate fossils, such as salamanders, feathered dinosaurs, pterosaurs and mammals (*Sullivan et al., 2014*). The most important localities that yield the Yanliao Biota are Daohugou in Ningcheng County of southeast Inner Mongolia (Haifanggou Fm.), Linglongta of Jianchang County of western Liaoning Province (Tiaojishan Fm.), and Mutoudeng of Qinglong County of northern Hebei Province (Tiaojishan Fm.; *Lü et al., 2013*; *Huang, 2015*, *2016*). From the Haifanggou Formation at Daohugou (*Liu et al., 2012*), pterosaurs are relatively rare, with *Jeholopterus ningchengensis* (*Wang et al., 2002*), *Pterorhynchus wellnhoferi* (*Czerkas & Ji, 2002*) and *Daohugoupterus delicatus* (*Cheng et al., 2015*). From the slightly younger Tiaojishan Formation at the Linglongta locality, pterosaurs are abundant in number and in diversity (see *Sullivan et al., 2014* for a review), with wukongopterids (*Wang et al., 2009*, *2010*; *Lü et al., 2009*; *Cheng et al., 2017*), *Jianchangopterus* (*Lü & Bo, 2011*), *Jianchangnathus* (*Cheng et al., 2012*) and *Fenghuangopterus* (*Lü, Fucha & Chen, 2010*). From the Tiaojishan Formation at Mutoudeng come *Luopterus mutoudengensis* (*Lü & Hone, 2012*; *Hone, 2020*), *Qinglongpterus guoi* (*Lü et al., 2012*) and *Changchengopterus pani* (*Lü, 2009*). It is from the Mutoudeng locality that comes the new specimen herein described (Fig. 1).

The Tiaojishan Formation is mainly distributed in the Chengde Basin (Maoniujiao–Xiaoguozhangzi–Jiyuqing Area) in northern Hebei Province. It is around 300 m thick (*Zhang & Chen, 2015*). It is mainly composed of neutral volcanic rock (*Zhang & Chen, 2015*). The lithology of the lower member includes dark grey, grey purple trachyandesites, quartz trachyandesites, small trachyandesitic agglomerate, small trachyandesitic ignimbrite (*Zhang & Chen, 2015*). The lithology of upper member includes dark grey, burgundy trachyandesites, trachyandesitic agglomerate, partially containing

grayish purple, grayish green sedimentary tuff, tuffaceous conglomerate and tuffaceous sandstone (*Zhang & Chen, 2015*).

*Zhang, Wang & Liu (2008)* analyzed samples of volcanic rock from several typical localities (Luanping Basin, Chengde Basin, Sanshijiazi Basin and Jinlingsi-Yangshan Basin), utilizing LA-ICP-MS Zircon U-Pb. Their result suggest that the lower limit age of the Tiaojishan Formation should be around 165 Ma. *Li et al. (2019)* analyzed samples of volcanic rock from the bottom of the lower section and andesite from the top of the upper section, utilizing LA-ICP-MS Zircon U-Pb. Their result gave an age range of 170–153 Ma for the Formation as a whole, that is, from the Bajocian until the Kimmeridgian. A specific dating for the strata of the Linglongta locality has been provided by *Liu et al. (2012)*, in order to provide a constrained age range for Linglongta wukongopterid pterosaurs. The bottom and the top of this locality were dated, resulting in an age range of 161–160 Ma (*Liu et al., 2012*), falling within the Oxfordian (early Late Jurassic). Specific dating under geochemical approaches still lack for the Mutoudeng locality. However, biostratigraphic studies, based mainly on conchostracans, suggest that the Linglongta and Mutoudeng strata are chronocorrelate (*Chu et al., 2016*).

# MATERIALS AND METHODS

## Computed tomography scanning

JPM-2012-001 was computed tomography (CT) scanned using a Nikon XTH225ST scanner at the Laboratory of Stratigraphy and Paleontology, Institute of Geology, Chinese Academy of Geological Sciences (IG-CAGS), Beijing, China. The specimen was scanned at 160 kV and 131 µA. The data set includes 2,000 image slices (2,000 × 2,000 pixels) with a slice thickness of 0.121 mm. The data was imported into digital visualization software Avizo (version 9.1) for image processing and visualization.

## Phylogenetic analysis

Concerning terminal taxa, our phylogenetic analysis is focused on non-pterodactyloid pterosaurs, following previous works that also focused on these forms (e.g. *Dalla Vecchia, 2009*, *2019*; *Andres, Clark & Xing, 2010*; *Lü et al., 2012*). Concerning our character list, we have gathered discrete characters from *Vidovic & Martill (2017)*, *Longrich, Martill & Andres (2018)* and *Dalla Vecchia (2019)*, all of which further encompass data from previous studies (e.g. *Kellner, 2003*; *Unwin, 2003*; *Dalla Vecchia, 2009*; *Lü et al., 2009*; *Wang et al., 2012*; *Naish, Simpson & Dyke, 2013*; *Andres, Clark & Xu, 2014*; *Britt et al., 2018*). Following previous works, we did not employ composite coding (*Colless, 1985*). The character list is available in Supplemental File 1 (a nexus format file for the software Mesquite, containing the data matrix) and Supplemental File 2 (a TNT file ready for executing the analysis, that can also be opened as a txt file).

We did not employ the treatment of continuous data as such (for discussions on the subject see *Goloboff, Mattoni & Quinteros, 2006*; *Bardin et al., 2014*; *Mongiardino Koch, Soto & Ramírez, 2015*; *Vidovic, 2018*). The original discretized quantitative characters from previous analyses (see our character list) were not modified, except for morphometric characters 270 (humerus/femur length, modified from *Kellner (2003)*) and 368

(tibia/femur length). Discrete states for the morphometric characters 270 and 368 were categorized (discretized) by using the gap-weighting method (*Thiele, 1993*). In order to optimize the phylogenetic signal, following *Bardin et al. (2014)*, state number was set at 3. The morphometric dataset subjected to gap-weighting is available as Table S1. The resulting categorization is presented in Table S1 and the data matrix (Supplemental Files 1 and 2). Quantitative characters 1, 45, 106, 152, 191, 264, 265, 270, 285, 289, 290, 293, 304, 313, 320, 321, 323 and 362 were treated as ordered. Following other works, all characters were equally weighted (e.g. *Fitzhugh, 2006*).

The analysis was performed using TNT (*Goloboff, Torres & Arias, 2018*) and was divided in two steps. The first search was performed using New Technology Search (using Sectorial Search, Ratchet, Drift and Tree fusing, default parameters), with random seed = 0. Subsequently, using trees from RAM, we performed a Traditional Search swapping (using TBR, collapsing trees after search). The TNT file is available as Supplemental File 2.

### Nomenclatural acts

The electronic version of this article in Portable Document Format will represent a published work according to the International Commission on Zoological Nomenclature (ICZN), and hence the new names contained in the electronic version are effectively published under that Code from the electronic edition alone. This published work and the nomenclatural acts it contains have been registered in ZooBank, the online registration system for the ICZN. The ZooBank Life Science Identifiers (LSIDs) can be resolved and the associated information viewed through any standard web browser by appending the LSID to the prefix http://zoobank.org/. The LSID for this publication is: urn:lsid:zoobank.org:pub:15997DEB-0EF7-40F6-80B0-2C40ED47D43B. LSID for the new genus: urn:lsid:zoobank.org:act:C1268C7D-80AA-4854-93E7-0E60220A05BC. LSID for the new species: urn:lsid:zoobank.org:act:048E9ADE-8C3A-47D4-B074-DCEFA40BDE9A. The online version of this work is archived and available from the following digital repositories: PeerJ, PubMed Central and CLOCKSS.

## RESULTS

**Systematic Paleontology**

Pterosauria Owen, 1842

Novialoidea *Kellner, 2003*

Breviquartossa *Unwin, 2003*

Monofenestrata *Lü et al., 2009*

Anurognathidae Kuhn, 1937

**Batrachognathinae *Kellner et al., 2010***

**Definition.** The most inclusive clade containing *Batrachognathus volans* but not *Anurognathus ammoni* (*Kellner et al., 2010*).

**Synapomorphies.** Humeral deltopectoral crest reduced (less wide than humeral shaft; and less wide than proximodistally long), humeral deltopectoral crest subrectangular, ulnar

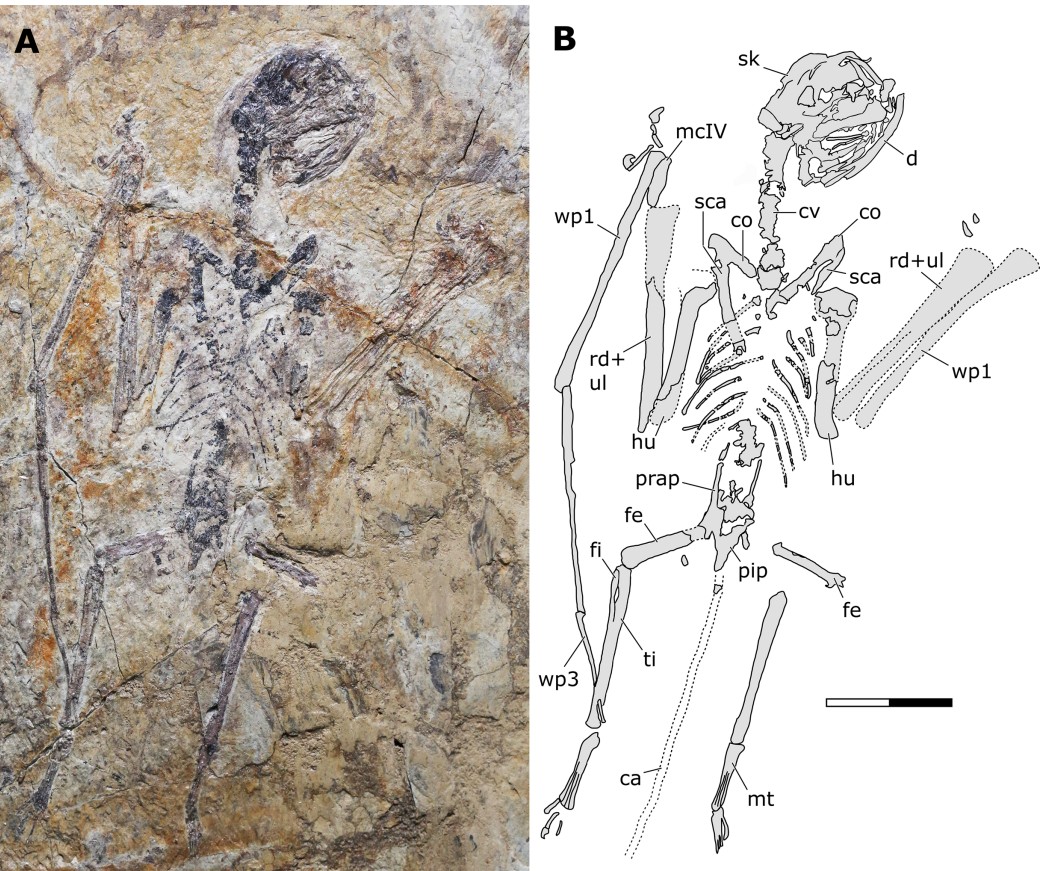

**Figure 2** *Sinomacrops bondei* **tax. nov., holotype (JPM-2012-001) overview.** (A) Photograph; and (B) schematic drawing. Abbreviations: ca, caudal vertebrae; co, coracoid; cv, cervical vertebrae; d, dentary; fe, femur; fi, fibula; hu, humerus; mcIV, metacarpal IV; pip, puboischiadic plate; prap, preacetabular process of the illium; rd, radius; sca, scapula; sk, skull; ul, ulna; wp, wing phalanx. Scale bar equals 20 mm.     

crest of humerus rounded, humeral/femoral length ratio over 1.60, tibial/femoral length ratio over 1.70.

**Included species.** *Batrachognathus volans* and *Sinomacrops bondei* gen. et sp. nov.

*Sinomacrops bondei* gen. et sp. nov.

**Etymology.** The generic name is a combination of *Sino*, *macro* and *ops*; which are Ancient Greek for China, large, and eyes/face, respectively. This is in reference to both the large eyes and the broad faces that are typical of anurognathids, and to the Chinese origin of the new species. The specific epithet honors paleontologist Niels Bonde, for his many scientific contributions and being an inspiration for us.

**Holotype.** JPM-2012-001 (Figs. 2–6).

**Locality and horizon.** Mutoudeng, Qinglong County of Hebei Province. Daohugou Beds (Callovian-Oxfordian 164-158 Ma) of the Tiaojishan Formation (see *Liu, Zhao & Liu 2006*; *Liu et al., 2006*; *Gao & Shubin, 2012*).

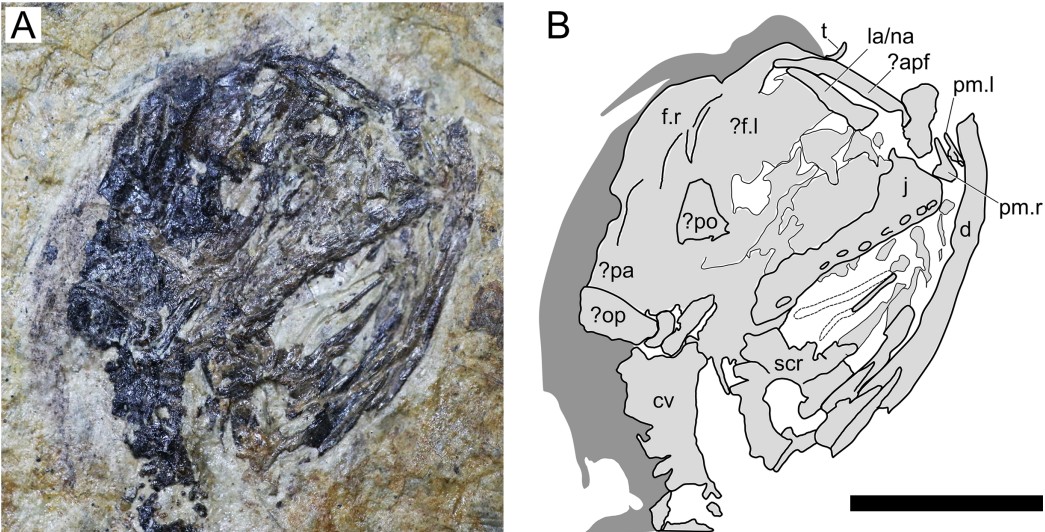

**Figure 3** ***Sinomacrops bondei* tax. nov., skull of JPM-2012-001.** (A) Photograph; and (B) schematic drawing. Light grey represents bones; dark grey represents soft tissue. Abbreviations: apf, anterior process of the frontal; cv, cervical vertebrae; d, dentary; f, frontal; j, jugal; la, lacrimal; na, nasal; pa, parietal; po, postorbital; pm, premaxilla; op, opisthotic; scr, sclerotic ring. Scale bar equals 10 mm.

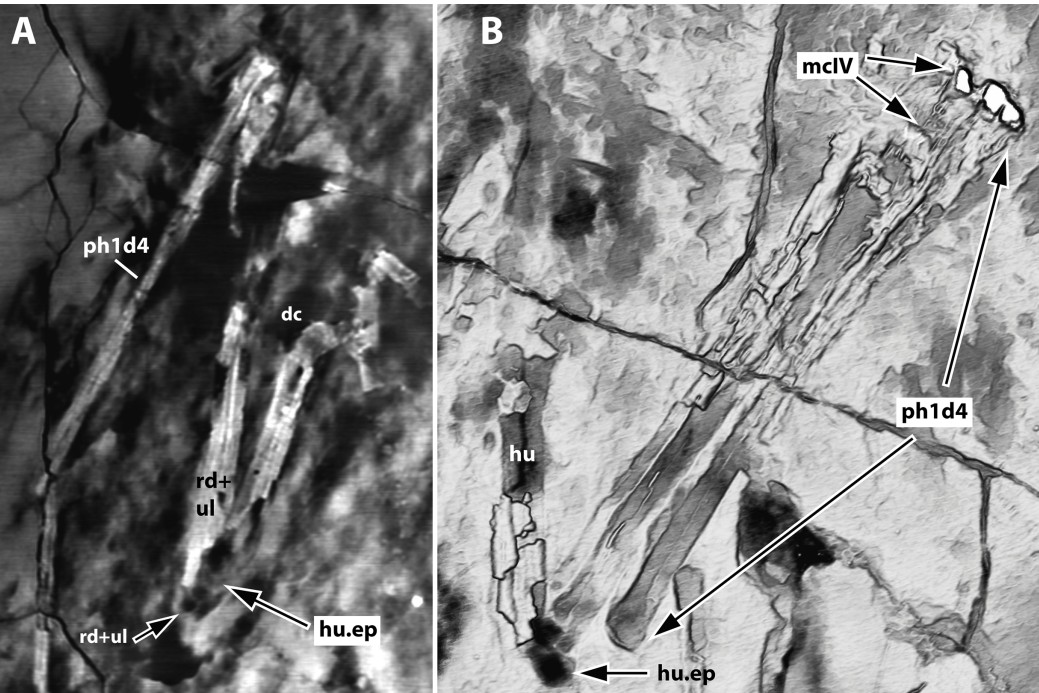

**Figure 4 Computed-tomography images of the wings of JPM-2012-001.** (A) Right wing; (B) left wing. Abbreviations: d, digit; dc, deltopectoral crest; hu.ep, humeral epiphysis; mc, metacarpal; ph, phalanx; rd, radius; ul, ulna.

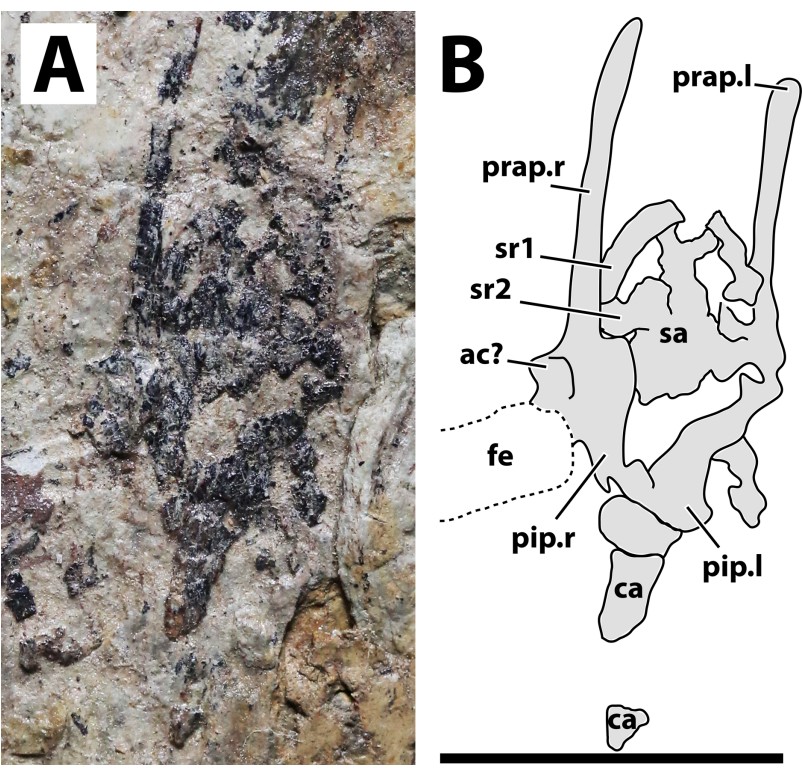

**Figure 5 Sacral region of JPM-2012-001.** (A) Photograph; (B) schematic drawing. Abbreviations: ac, acetabulum; ca, caudal vertebrae; fe, femur; pip, puboischiadic plate; prap, preacetabular process of the illium; sa, sacral vertebrae; sr, sacral rib. Scale bar equals 10 mm.

**Diagnosis.** The new taxon exhibits two autapomorphies: first three maxillary alveoli closely spaced, and tibiotarsus twice as long as the femur.

**Description**

**Generalities.** JPM-2012-001 comprises a crushed skeleton (Fig. 2). While the cranium and some cervical vertebrae are exposed in right lateral aspect (Fig. 3), the remaining of the skeleton is exposed in ventral view. The preserved bone tissue exhibits a fragile, brittle condition. In consequence, in many regions of the skeleton, fragments of bone tissue have been lost posterior to collection of the specimen. These lost fragments left clear impressions on the matrix, indicating where they were originally present. Lost fragments include mainly the caudal vertebrae, sternum, distal epiphysis of right humerus, proximal epiphyses of right ulna and radius, parts of the left humerus, and most of the left manus.

Micro CT scan resulted in images with only limited resolution. Nonetheless, the images permitted better visualization of some impressions on the matrix (represented by empty spaces on the slices), helping in the identification of some bone limits and extensions. Such was the case of elements of the left wing (humerus epiphysis, radius and ulna, wing metacarpal and first wing phalanx), as well as the right humerus (Fig. 4). CT images did not provide enough resolution for additional data on other skeletal regions.

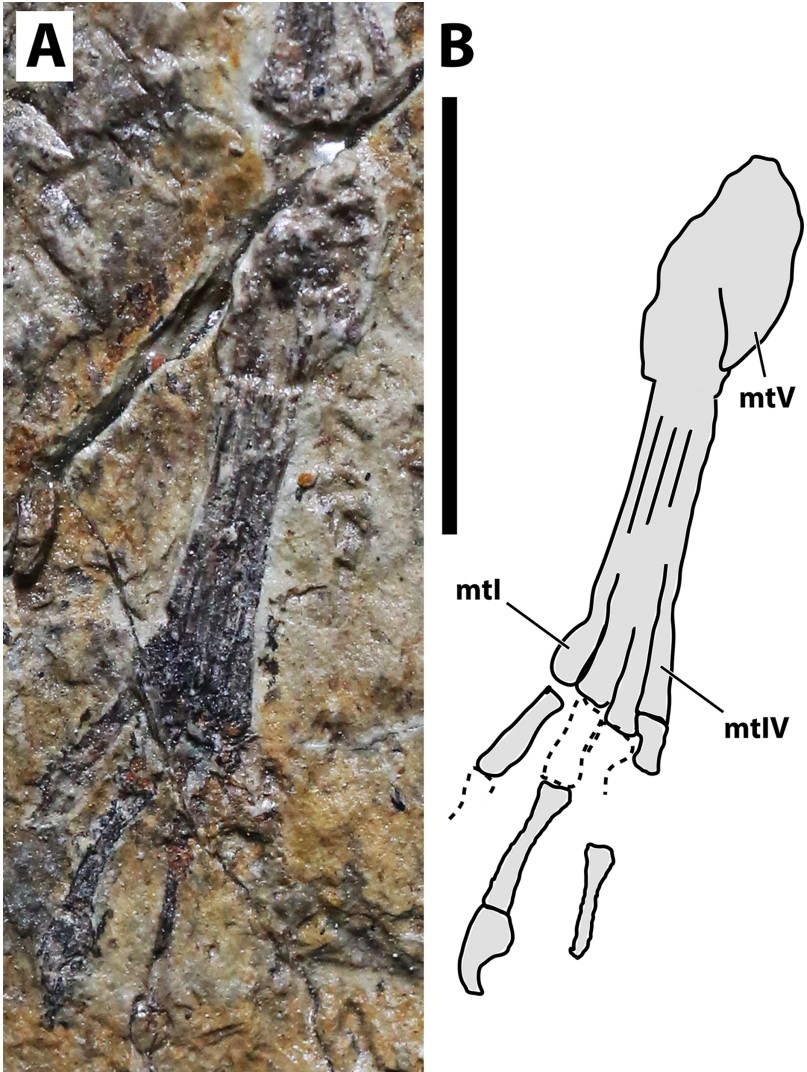

**Figure 6 Right pes of JPM-2012-001.** (A) Photograph; (B) schematic drawing. Abbreviations: mt, metatarsal. Scale bar equals 10 mm.

**Soft tissue.** The skeleton includes preservation of soft tissue patches. The dorsal margin of the skull is covered by skin impressions that descends onto the neck region (Fig. 3). An irregular patch of soft tissue lateral to the left tibiotarsus suggests that the brachiopatagium extended posteriorly onto the distal region of the crus. A brachiopatagium extending distally on the crus is consistent with what is seen in *Jeholopterus ningchengensis* (see *Kellner et al., 2010*) and pterosaurs in general (see *Elgin, Hone & Frey, 2011*). Another large patch of soft tissue is present medial to the right hindlimb, extending from the femoral region until the distal fifth of the tibiotarsus. This implies in an extensive cruropatagium, though it is not clear if the tail is integrated with it. An extensive cruropatagium can also be found in *Sordes pilosus* (*Unwin & Bakhurina, 1994*). Deeper investigation of the soft tissue remains of JPM-2012-001 is beyond the scope of the present contribution and shall be presented elsewhere.

**Cranium.** The cranium of JPM-2012-001 is exposed in right lateral aspect (Fig. 3). A small pair of bones on the rostral tip of the skull seem to represent an unfused pair of premaxillae. Individually, they comprise basically two processes, one ascending and another one extending posteriorly. This indicates that the fused premaxillae would display a T-shape similar to other anurognathids, as seen in *Batrachognathus volans* (*Riabinin, 1948*) and *Anurognathus* (*Bennett, 2007*). The right premaxilla is exposed laterally, while the left one is slightly displaced and exposed in anteromedial aspect. The dorsal process of the premaxilla seems to have extended for no further than half the height of the skull. It contacts an anterior process of the frontal, which is elongated and thin, as in *Anurognathus ammoni* (see *Bennett, 2007*). The posterior process of the premaxillae participates on the occlusal jaw margin, and presumably contacted the maxillae, though the bones are slightly displaced and not in natural contact.

The maxilla and jugal are fused, with not visible sutures, forming a large bony structure, posterior to the premaxillae. It forms most of the jaw as well as the ventral border of the orbit. The jugo-maxilla structure houses 9 alveoli. The lacrimal process of the jugal is present on the anterior region of this structure. It forms the anteroventral border of the orbit, and the posteroventral margin of the nasoantorbital fenestra. It is incomplete dorsally, but is clearly slender, much higher than long. The nasal and the lacrimal cannot be distinguished.

It appears that both frontals are visible: the right one in lateral aspect, and the left one in medial aspect. They are both positioned on the posterodorsal region of the orbit, and take part in the dorsal margin of the skull itself. Their limits are not clear, but the dorsal margin of the right frontal is convex, as is the dorsal margin of the skull in lateral view. Posterior to the right frontal, two bones are tentatively interpreted as the right parietal and a misplaced right opisthotic.

A large bone bearing 9 alveoli forms most of the right upper jaw margin, and is here interpreted as a jugomaxilla complex, similar to the one reported for *Anurognathus ammoni* where the jugal overlays the maxilla laterally, fusing with it (*Bennett, 2007*). The structure is seen in lateral view, and no sutures can be seen separating jugal from maxilla. The right jugomaxilla seems to be disarticulated from both the quadrate and the premaxilla.

A triangular bone located on the posterior margin of the orbit is tentatively interpreted as the postorbital. If this identification is correct, then the postorbital of *Sinomacrops* is quite different from that of *Anurognathus*, which is very slender (and dorsoventrally elongated). Thus, the postorbital of *Sinomacrops* would be more similar to that of some non-anurognathid pterosaurs such as *Dimorphodon* or rhamphorhynchids (e.g. *Padian, 1983*; *Wellnhofer, 1991*).

Ventral to the jugomaxilla, a rod-like bone is preserved, adjacent to the impression of another similar rod-like bone. These two rod-like bones are interpreted as either members of the hyoid apparatus, or members of the palate, which is composed of rod-like bones and bony processes (pterygoids, palatines, vomer, ectopterygoids) in *Anurognathus ammoni*, *Jeholopterus ningchengensis* and *Batrachognathus volans* (*Riabinin, 1948*; *Bennett, 2007*; *Yang et al., 2019*).

A partial sclerotic ring is preserved, displaced from its natural position and located ventral to the posterior region of the skull. Though partially preserved, it is complete enough to allow for an estimation of its diameter. It is estimated as ~7 mm, what is close to the estimated diameter of the orbit (7.5 mm).

**Mandible.** An hemimandible is exposed beneath the skull (Fig. 3). No alveoli can be observed, suggesting that it is the left hemimandible in ventral view. We infer that this hemimandible is complete because its length equals that of the upper jaw. It is only slightly bowed, as in *Batrachognathus volans*, instead of strongly semicircular as in the jaws of *Dendrorhynchoides* (*Ji & Ji, 1998*), *Luopterus* (*Lü & Hone, 2012*; *Hone, 2020*), *Jeholopterus* (*Wang et al., 2002*), *Anurognathus* (*Bennett, 2007*) or *Vesperopterylus* (*Lü et al., 2018*).

**Dentition.** A single preserved tooth crown is visible, displaced from the jaws and located near the anterodorsal region of the skull (Fig. 3). This tooth is slender and slightly recurved. At least 9 alveoli are present on the right maxilla. The alveoli on the right premaxilla are unclear. The first three maxillary alveoli are closely spaced, with the spacing between them being shorter than their diameter. Posteriorly, the spacing between the subsequent alveoli is subequal to their diameter.

**Axial postcranium.** Throughout the whole specimen, the vertebrae are highly damaged and details of their anatomy cannot be retrieved (Fig. 2). Still, as the skeleton is almost complete, the lengths of each segment can be estimated, with 23 mm for the cervical series; 30 mm for the dorsal series; 11 mm for the sacral series; and >36 mm for the caudal series. The sacral series thus seems to have been elongated, similarly to the condition seen in the possible anurognathid *Mesadactylus* (see *Jensen & Padian, 1989*). The rib of the first sacral is strongly inclined posteriorly, while the rib of the second sacral is less inclined (Fig. 5). This configuration is very similar to that of *Mesadactylus* (see *Jensen & Padian, 1989*). At least 9 pairs of ribs anterior to the sacral region can be seen (Fig. 2), all of which are long and slender, and interpreted as dorsal ribs. This is the same number of dorsal ribs seen in *Dendrorhynchoides* (*Ji & Ji, 1998*), *Anurognathus* (*Bennett, 2007*) and *Jeholopterus* (*Wang et al., 2002*). Concerning caudal vertebrae, only three incomplete remains of proximal caudal centra are present, near the sacral region. They are simple, lacking lateral processes.

**Forelimb.** The scapulae and coracoids of JPM-2012-001 are elongate and slender, as in other anurognathids (e.g. *Bennett, 2007*; *Lü et al., 2018*). Although fragments of the bone tissue have been lost post-collection due to the brittle nature of the fossil, the remaining impression of the right humerus is quite clear upon close inspection. The deltopectoral crest is subrectangular, as can be better seen on the left side (Fig. 2). As in *Batrachognathus volans*, the deltopectoral crest of the humerus in JPM-2012-001 was reduced (less wide than proximodistally long, and less wide than humeral shaft) and rectangular in shape. The shape of the ulnar crest is rounded, but it is proximodistally shorter than the deltopectoral crest, as in other anurognathids (*Döderlein, 1923*; *Riabinin, 1948*; *Ji & Ji, 1998*, *Wang et al., 2002*; *Bennett, 2007*; *Lü & Hone, 2012*; *Lü et al., 2018*; *Yang et al., 2019*).

**Table 1 Measurements of JPM-2012-001.**

| Element | Right | Left |
|---|---|---|
| Scapula | ~1.95 | ? |
| Coracoid | ? | ~1.37 |
| Humerus | 2.36 | 2.39 |
| Radius/ulna | 3.63 | 3.47 |
| Metacarpal IV | ~0.67 | – |
| Wing phalanx 1 | 4.12 | ~3.84 |
| Wing phalanx 2 | 3.60 | – |
| Wing phalanx 3 | 1.81 | – |
| Femur | 1.36 | 1.31 |
| Tibiotarsus | 2.66 | 2.53* |
| Metatarsus | ~1.1 | ~1 |

Note:
Measurements are given in centimeters. Values for long bones correspond to their lengths. Interrogations mean the element is too incomplete for an informative value. Dashes mean the element is not preserved. Asterisk means the element is slightly incomplete.

Incomplete preservation prevents the observation of any details of ulna and radius, although their lengths can be assessed due to their clear impressions on both sides. The right wing-finger preserves complete first, second and third wing phalanges (Fig. 2). The distal region of the third wing phalanx underlies the tibia on the matrix, but the distal end can be seen due to damage on the tibia, revealing the phalanx beneath. The distal end of the third wing phalanx seems to be slightly expanded, indicating a probable articular region for a fourth phalanx, which is not preserved. A free digit with a long, slender proximal phalanx and a robust, strongly recurved ungual is preserved.

**Hindlimb.** Neither femora are fully preserved in terms of bone tissue, though impressions of the lost regions remain on both sides so that their total lengths can be confidently measured (Fig. 2). The right femur is preserved in an approximately natural position relative to the pelvic region, and only part of the proximal region was lost, though an impression remains, showing that it was preserved in articulation with the pelvis. The left femur is displaced, but the proximal region is preserved. The distal region is lost, but an impression also remains. The tibia is quite elongate relative to the femur (Fig. 2), more so than in any other anurognathid (Table 1). On the right crus, tibia and fibula are incompletely ossified, and a gap can be seen between the two (Fig. 2). Despite damage on the proximal region of the right metatarsus, the distal region is well-preserved. It can be clearly seen that the metatarsal IV is shorter than metatarsals II and III (Fig. 5). A single ungual can be identified on the right pes, which is slightly less robust than the manual unguals (Fig. 6).

**Ontogeny.** Specimen JPM-2012-001 has not reached osteological maturity, as indicated by the incomplete degree of fusion of the skull bones. The scapula and coracoid seem to be fused, although it remains unclear. Fusion of the extensor tendon process of the first wing phalanx is unclear, as the proximal region of this bone is not well-preserved. A fused

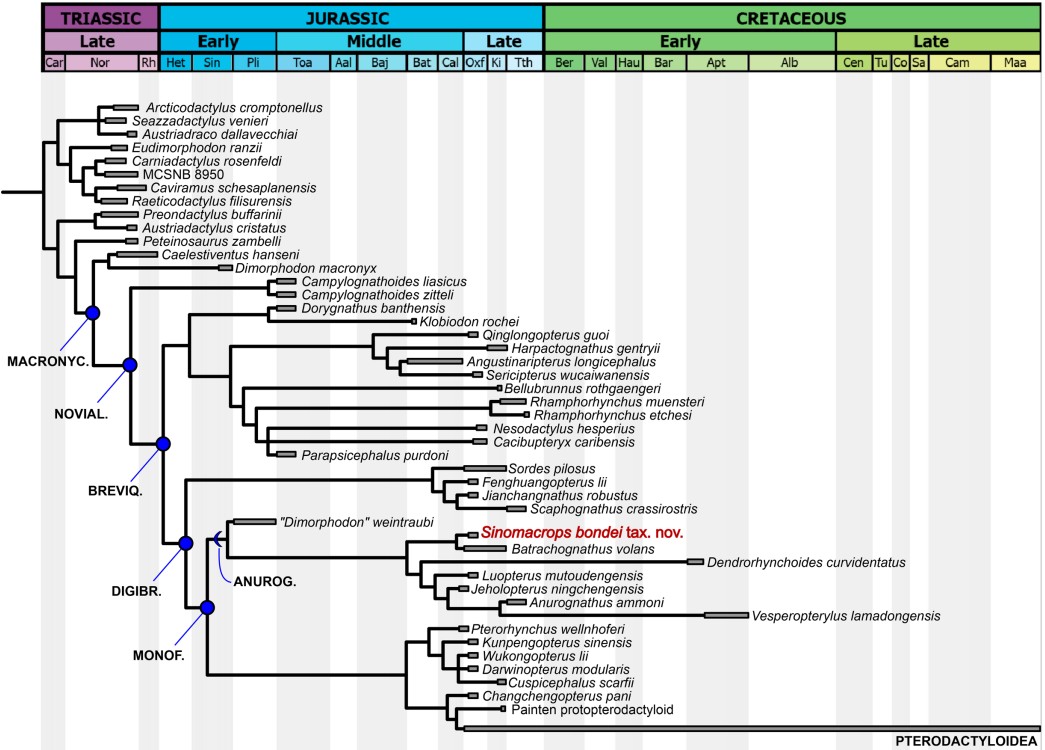

**Figure 7 Phylogenetic analysis results.** Strict consensus tree showing the phylogenetic relationships of *Sinomacrops bondei* and anurognathids. Dashed line indicates result exclusive to the semi-strict con-sensus tree.

puboischiadic plate is present, indicating the specimen must have reached at least "ontogenetic stage 4" of *Kellner (2015)* and is thus considered a subadult.

### Phylogenetic analysis results

Our analysis produced two most parsimonious trees, with 1,115 steps, CI of 0.456 and RI of 0.668. In the strict consensus tree (Fig. 7), the new species is the sister-group of *Batrachognathus volans*. The Anurognathinae were recovered with *Dendrorhynchoides* at the base, plus the newly recognized clade *Luopterus* + (*Jeholopterus* + (*Anurognathus* + *Vesperopterylus*)).

As in the results from *Dalla Vecchia (2019)*, "*Dimorphodon weintraubi*" is placed as the sister-group of a clade containing all other anurognathids. Under the branch-based definition of the Anurognathidae given by *Hone (2020)*, "*D. weintraubi*" can be considered as a basal anurognathid. For the first time, the Anurognathidae is recovered as the sister group of Darwinoptera + Pterodactyloidea. The synapomorphies are discussed further below.

## DISCUSSION

### Comparisons with other anurognathids

As detailed above, JPM-2012-001 exhibits a particular feature regarding its dentition: the first three maxillary alveoli are closely spaced, with the spacing between them being shorter

**Table 2 Comparative table showing skeletal element ratios among anurognathids.**

| Anurognathidae | hu/mcIV | hu/fe | hu/ul | hu+ul/fe+ti | ul/mcIV | ul/fe | sc/co | ph1d4/ul+mcIV | ph1d4/ti | ph2d4/ph1d4 | ph3d4/ph1d4 | ph3d4/ph2d4 | ph4d4/ph1d4 | fe/mcV | ti/fe | mtIII/ti | caS/fe |
|---|---|---|---|---|---|---|---|---|---|---|---|---|---|---|---|---|---|
| *Anurognathus ammoni* (holotype) | 2.91 | 1.19 | 0.70 | 1.16 | 4.18 | 1.70 | ? | 1.01 | 1.49 | ? | ? | ? | ? | 2.45 | 1.44 | 0.46 | 0.50 |
| *Anurognathus ammoni* (referred) | 3.64 | 1.25 | 0.70 | 1.26 | 5.10 | 1.76 | ? | 0.95 | 1.44 | 0.77 | 0.44 | 0.56 | ? | 2.90 | 1.39 | 0.42 | ? |
| *Vesperopterylus lamadongensis* | 2.75 | 1.35 | 0.74 | 1.34 | 3.73 | 1.83 | 0.97 | 0.96 | 1.64 | 0.81 | 0.60 | 0.74 | 0.12 | 2.04 | 1.37 | 0.47 | 0.59 |
| *Jeholopterus ningchengensis* (holotype) | 3.26 | 1.55 | 0.70 | 1.67 | 4.68 | 2.22 | 1.96 | 0.86 | 1.86 | 0.88 | 0.65 | 0.73 | 0.17 | 2.10 | 1.25 | 0.44 | ? |
| *Jeholopterus ningchengensis* (CAGS IG 02-81) | 3.39 | 1.52 | 0.78 | 1.59 | 4.03 | 1.99 | 1.28 | 0.88 | 1.88 | 0.89 | ? | ? | ? | 2.02 | 1.22 | 0.47 | ? |
| *Dendrorhynchoides curvidentatus* | 2.99 | 1.43 | 0.78 | 1.37 | 3.82 | 1.82 | 1.15 | 0.99 | 1.66 | 0.80 | ? | ? | ? | 2.4 | 1.37 | 0.45 | ? |
| *Luopterus mutoudengensis* holotype | 2.45 | 1.28 | 0.64 | 1.44 | 3.81 | 2.00 | 1.88 | 0.94 | 1.85 | 0.82 | 0.50 | 0.61 | 0.10 | 1.91 | 1.29 | 0.44 | >0.86 |
| NJU–57003 | 2.60 | 1.34 | 0.60 | 1.42 | 4.31 | 2.15 | 1.27 | 0.90 | 1.63 | 0.86 | 0.40 | 0.46 | 0.10 | 1.97 | 1.47 | 0.45 | 1.78 |
| IVPP V16728 | ? | 1.43 | ? | ? | ? | ? | ? | ? | ? | ? | ? | ? | ? | ? | ~1.40 | 0.38 | >1.49 |
| *Sinomacrops bondei* | 3.55 | 1.77 | 0.66 | 1.51 | 5.29 | 2.70 | 1.42 | 0.97 | 1.59 | 0.87 | 0.44 | 0.50 | ? | ~2 | 1.99 | 0.48 | >1.69 |
| *Batrachognathus volans* | ? | 1.93 | ? | ? | ? | ? | ? | ? | ? | ? | ? | ? | ? | ? | 1.75 | ? | 1.47* |

**Note:**
The asterisk indicates a value taken from the referred specimen of *Batrachognathus volans* (*Costa et al., 2013*). The other values for this species were taken from the holotype (*Riabinin, 1948*).

than their diameter; while the spacing between the subsequent alveoli is subequal to their diameter. This pattern is unprecedented for anurognathids. In *Batrachognathus volans*, *Dendrorhynchoides curvidentatus*, *Jeholopterus ningchengensis* and *Anurognathus ammoni*, tooth spacing is constant and larger than tooth diameter (*Riabinin, 1948*; *Ji & Ji, 1998*; *Ji & Yuan, 2002*; *Bennett, 2007*). In *Vesperopterylus lamadongensis*, tooth spacing is also constant, and subequal to (only fractionally larger than) tooth diameter (*Lü et al., 2018*). The pattern of tooth spacing in *Luopterus mutoudengensis* is so far unclear (*Lü & Hone, 2012*; *Hone, 2020*).

Another particular feature is its tibiotarsus/femur length ratio, which is unique within anurognathids (and pterosaurs overall) in that the tibiotarsus is about twice as long as the femur (Table 2; Table S1). In *Batrachognathus volans*, this same ratio is 1.75, while it ranges from 1.22 to 1.47 in other anurognathids (Table 2).

Apart from the unique features mentioned above, *Sinomacrops bondei* further differs from *Batrachognathus volans* in exhibiting a relatively larger ulnar crest of the humerus (*Riabinin, 1948*; *Hone, 2020*). The new species further differs from *Anurognathus*, *Jeholopterus* and *Vesperopterylus* in humerus deltopectoral crest shape (trapezoidal in
the latter three taxa) and in exhibiting an elongate tail, longer than the dorsal series (*Hone, 2020*). The new species also differs from *Luopterus mutoudengensis* and *Dendrorhynchoides curvidentatus* in the morphology of the deltopectoral crest of the humerus, which is relatively larger and triangular in shape in the latter two (*Ji & Ji, 1998*; *Lü & Hone, 2012*; *Hone, 2020*).

## Diversity of the Anurognathidae

It has been observed that some aspects of anurognathid morphology did not change from the Middle Jurassic (in the form of *Jeholopterus*) to the Early Cretaceous (in the form of *Dendrorhynchoides*; prior to the description of the even younger *Vesperopterylus*), such as skull shape, palate morphology and dentition (*Unwin, Lü & Bakhurina, 2000*; *Bennett, 2007*). This has led to the conclusion that the anurognathid bauplan was rather conservative (*Unwin, Lü & Bakhurina, 2000*; *Bennett, 2007*). Nonetheless, several features of anurognathid morphology exhibit some variation, what has been relatively poorly explored so far (*Hone, 2020*).

Concerning the particular shape of the anurognathid jaw in dorsal/ventral views, we note that there exists some variation. The roundness of the jaws (both upper and lower) is relatively more pronounced in anurognathines, as can be seen particularly in *Anurognathus* (*Bennett, 2007*), *Jeholopterus* (*Wang et al., 2002*; *Ji & Yuan, 2002*), *Vesperopterylus* (*Lü et al., 2018*) and NJU–57003 (*Yang et al., 2019*). In these, the arching of the jaws is abrupt and approximately continuous (Fig. 8), describing a semicircular shape (Fig. 9). In contrast, in *Batrachognathus* and *Sinomacrops*, the arching of the jaws is less pronounced and relatively more gradual (Fig. 8), making the jaws rather elliptical instead of semicircular (Fig. 9).

Some variation on tooth morphology is also found within anurognathids. The dentition of *Anurognathus ammoni* is homodont and was referred to as pupiform, given their resemblance to dipteran pupae (*Bennett, 2007*). The only complete tooth preserved in the referred specimen of *Anurognathus ammoni* is short, has a subcylindrical base and tapers to a sharp end, being only slightly recurved (*Bennett, 2007*). This is very similar to the condition seen in *Vesperopterylus lamadongensis*, except that in this taxon the teeth are relatively stouter (see *Lü et al., 2018*). However, the teeth in *Jeholopterus ningchengensis*, NJU–57003, *Dendrorhynchoides curvidentatus* and *Batrachognathus volans* are relatively longer and more recurved. The single tooth visible in the holotype of *Sinomacrops bondei* is superficially similar to these latter taxa. *Luopterus mutoudengensis* is unique within anurognathids, having been described as exhibiting a heterodont dentition comprising slender, sharp teeth anteriorly and relatively more robust, short teeth posteriorly (*Lü & Hone, 2012*). However, recently, *Hone (2020)* suggested that the purported robust teeth may in fact be bone shards.

According to *Lü & Hone (2012)* and *Hone (2020)*, a noticeable amount of variation in anurognathids is also expressed through the shape of the deltopectoral crest of the humerus (Fig. 10), as follows: rounded for *Anurognathus ammoni*, alate for *Jeholopterus ningchengensis*, triangular for *Dendrorhynchoides curvidentatus* and *Luopterus mutoudengensis*, and sub-rectangular (or parallelogram shaped, *Hone, 2020*) for

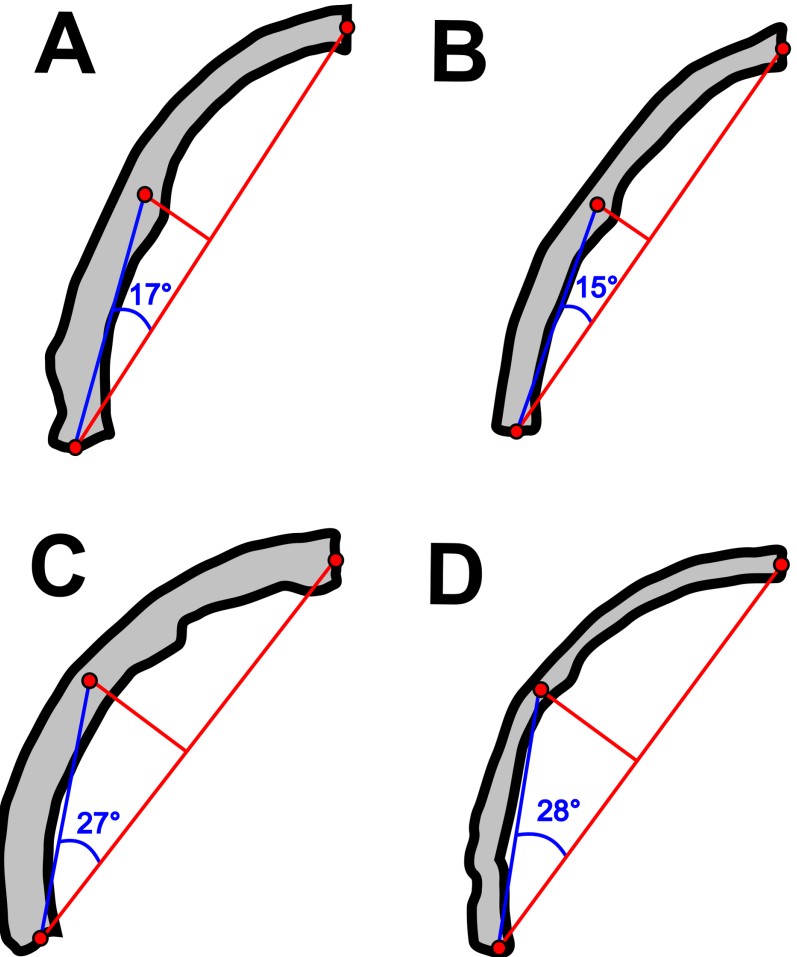

**Figure 8 Variation in the arching of the hemimandible in anurognathids.** Schematic drawings of anurognathid hemimandibles in ventral view. (A) *Batrachognathus volans* (based on *Riabinin, 1948*). (B) *Sinomacrops bondei*. (C) *Jeholopterus ningchengensis* (based on *Yang et al., 2019*). (D) *Vesperopterylus lamadongensis* (based on *Lü et al., 2018*). Not to scale, adjusted to matching sizes. The blue line connects the centroid and the posterior point of the hemimandible. The long red line connects the posterior and anterior points. The angle between these lines is higher in *Jeholopterus* and *Vesperopterylus*, corresponding to a higher arching degree of the jaws compared to *Batrachognathus* and *Sinomacrops*.

*Batrachognathus*. However, in the holotype of *Anurognathus*, the structure is not rounded, but trapezoidal (*Döderlein, 1923*; *Wellnhofer, 1991*). Despite not being clearly depicted as such in the line-drawings, the humeral deltopectoral crest of the second specimen of *Anurognathus* was also explicitly described as trapezoidal (see *Bennett, 2007*), and is probably relatively smaller due to allometric growth. In *Vesperopterylus*, the deltopectoral crest of the humerus is also trapezoidal, very similar in shape to *Anurognathus* (see *Lü et al., 2018*). In the North Korea specimen, the deltopectoral crest of the humerus seems to be trapezoidal as well (*Gao et al., 2009*). Furthermore, even though the "alate" condition seen in *Jeholopterus* is unique to it, it is still very similar to the trapezoidal conditions of *Anurognathus* and *Vesperopterylus*, differing only in being longer and more curved—they are thus all coded as "trapezoidal" in our analysis (see Supplemental Information).

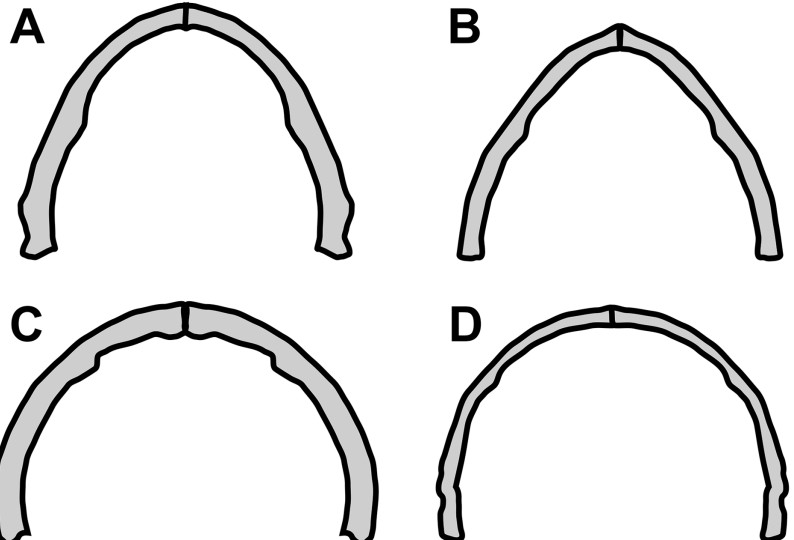

**Figure 9 Variation in anurognathid jaw shape.** Schematic drawings of anurognathid mandibles in ventral view. (A) *Batrachognathus volans* (based on *Riabinin, 1948*). (B) *Sinomacrops bondei*. (C) *Jeholopterus ningchengensis* (based on *Yang et al., 2019*). (D) *Vesperopterylus lamadongensis* (based on *Lü et al., 2018*). Not to scale, adjusted to matching sizes.  

Concerning other anurognathids, NJU–57003 is similar to *Dendrorhynchoides* and *Luopterus* in exhibiting a subtriangular deltopectoral crest of the humerus (*Yang et al., 2019*). In the holotype of *Sinomacrops bondei*, the impression of the deltopectoral crest of the humerus reveals it was subrectangular in shape, being similar to that of *Batrachognathus volans*, but different in that it is relatively shorter and that its distal margin is even straighter than in *B. volans* (Figs. 10A and 10B). *Sinomacrops* and *Batrachognathus* are further unique in exhibiting deltopectoral crests that are reduced in size, being less wide than the humeral shaft, and less wide than proximodistally long (Figs. 10A and 10B).

Still concerning the proximal region of the humerus, considerable variation can also be found in the shape of the ulnar crest. In *Batrachognathus volans* and *Sinomacrops bondei*, the distal margin of the ulnar crest is rounded (Figs. 10A and 10B). In *Dendrorhynchoides curvidentatus*, it is slightly more prominent, subtriangular (Fig. 10). In *Jeholopterus*, it is particularly reduced, and is also prominent (Fig. 10D). In *Anurognathus* and *Vesperopterylus*, it is relatively elongated and oriented obliquely to the humeral shaft (Figs. 10E and 10F).

Another interesting variation seen within anurognathids concerns the length of their caudal series and the morphology of their caudal vertebrae (*Lü & Hone, 2012*; *Costa et al., 2013*; *Jiang et al., 2014*). *Batrachognathus* and the indeterminate specimens IVPP V16728 and NJU–57003 exhibit the typical non-pterodactyloid condition, with long tails (longer than femur length) and caudal vertebrae bearing long filiform processes of the zygapohyses and haemapophyses (*Costa et al., 2013*; *Jiang et al., 2014*; *Yang et al., 2019*). *Luopterus mutoudengensis* exhibits a relatively short caudal series, that is shorter than the dorsal series and equals 0.85 the femur length (*Lü & Hone, 2012*). As for caudal vertebrae morphology, *Luopterus* was reported to bear filiform processes interpreted as

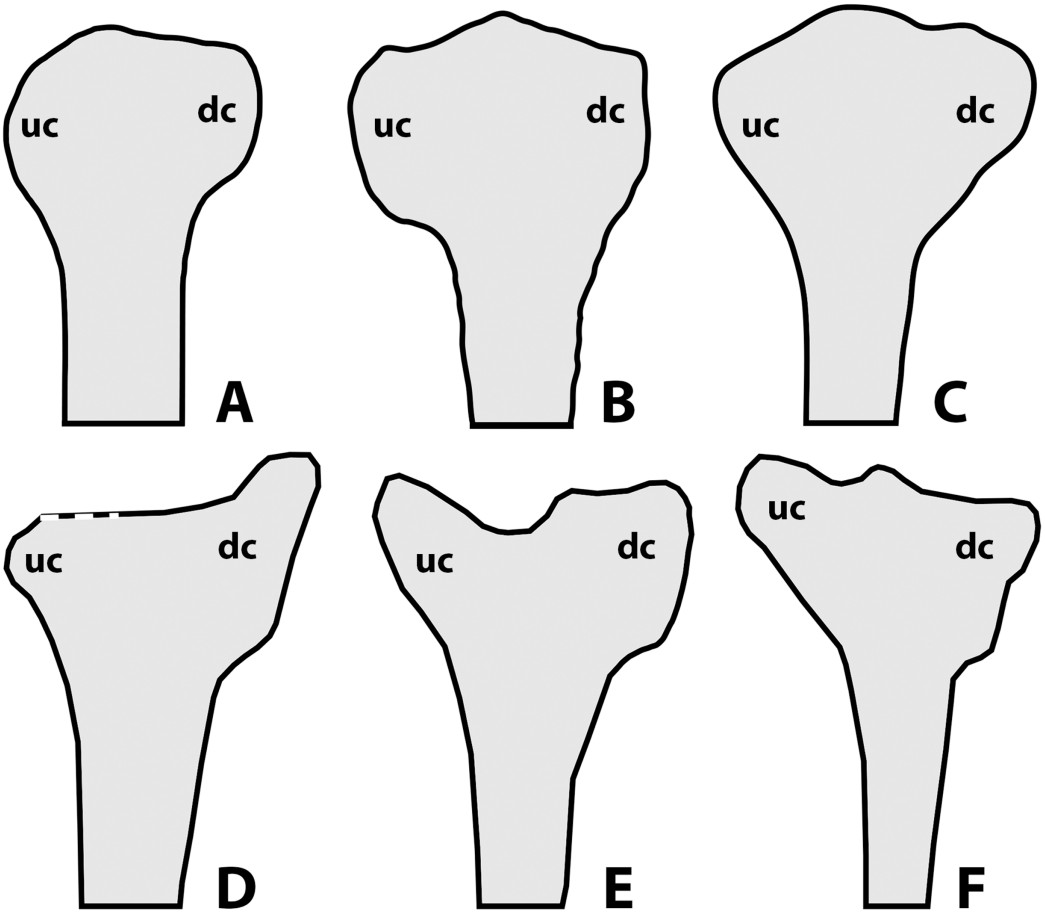

**Figure 10 Schematic drawings of anurognathid humeri.** (A) *Batrachognathus volans* (based on *Riabinin, 1948*). (B) *Sinomacrops bondei*. (C) *Dendrorhynchoides curvidentatus* (based on *Ji & Ji, 1998*). (D) *Jeholopterus ningchengensis* (based on *Kellner et al., 2010*). (E) *Vesperopterylus lamadongensis* (based on *Lü et al., 2018*). (F) *Anurognathus ammoni* based on *Wellnhofer (1991)*. Not to scale, adjusted to matching sizes. Abbreviations: dc, deltopectoral crest; uc, ulnar crest.

haemapophyses (*Lü & Hone, 2012*). *Jiang et al. (2014)* have suggested that *Luopterus mutoudengensis* possessed processes produced by both the zygapophyses and haemapophyses, and we agree this is rather likely. In our matrix, the haemapophyses processes are coded as present and the zygapophyses processes as "?" until a first-hand reassessment of the specimen is provided. In *Jeholopterus* (both specimens), the tail is most likely shorter than the femur, though details of vertebral morphology cannot be assessed (*Wang et al., 2002*; *Ji & Yuan, 2002*; *Jiang et al., 2014*; *Yang et al., 2019*). Finally, *Anurognathus* and *Vesperopterylus* possess quite shortened tails (accounting for under 60% the femur length) and caudal vertebrae without any filiform processes, in a homoplastic condition relative to the Pterodactyloidea (see *Jiang et al., 2014*). In *Sinomacrops bondei*, even though the total extent of the caudal series is uncertain, the preserved impression indicates it was longer than the femur—in fact, longer than the entire hindlimb.

## Intrarelationships of the Anurognathidae

Our phylogenetic analysis places *Sinomacrops bondei* alongside *Batrachognathus volans* forming the Batrachognathinae, separately from the clade containing all other Chinese anurognathids plus *Anurognathus ammoni* (the Anurognathinae as herein defined). Five synapomorphies support Batrachognathinae in our analysis: char. 269 (2), humeral/femoral length proportion (over 1.6); char. 271 (0) the width of the humeral deltopectoral crest (reduced, less wide than proximodistally long), char 272 (3), the shape of the deltopectoral crest (subrectangular); char. 280 (2), the shape of the ulnar crest of the humerus (rounded); and char. 367 (2), the tibia/femur length proportion (over 1.7).

The Anurognathinae would be composed of, according to our results, *Dendrorhynchoides curvidentatus*, *Luopterus mutoudengensis*, *Jeholopterus ningchengensis*, *Anurognathus ammnoni* and *Vesperopterylus lamadongensis*. These taxa share the following synapomorphies: char. 30 (2) the semicircular arching of the jaws, distinct from the elliptical one seen in batrachognathines, char. 244 (1) caudal series shorter than the dorsal series, char. 275 (1) deltopectoral crest subequal to humeral head in size and char. 310 (5) pteroid curved and subparallel-sided (*Andres, Clark & Xu, 2014*).

The non-monophyly of the genus *Dendrorhynchoides* encompassing *D. curvidentatus* plus *D. mutoudengensis* (*Lü & Hone, 2012*) is corroborated here, which is consistent with *Wu, Zhou & Andres (2017)* and *Hone (2020)*. *Luopterus mutoudengensis* is recovered as the sister-group of the *Jeholopterus–Anurognathus–Vesperopterylus* clade, with which it shares char. 378 (0), a straight last phalanx of pedal digit V (whereas this phalanx is curved in *Dendrorhynchoides curvidentatus*). The straight condition is a synapomorphy joining these taxa, while the curved condition is plesiomorphic for anurognathids and present at the base of the Novialoidea, as seen in *Campylognathoides*, "*Dimorphodon weintraubi*", *Changchengopterus pani* and wukongopterids (*Clark et al., 1998*; *Lü, 2009*; *Padian, 2008a*, *2008b*; *Wang et al., 2009*, *2010*).

The clade composed of *Jeholopterus ningchengensis*, *Anurognathus ammoni* and *Vesperopterylus lamadongensis* is supported by three synapomorphies: char. 272 (1) deltopectoral crest of the humerus trapezoidal and broad, char 241 (0) caudal vertebrae lacking filiform zygapophyses, and char. 242 (0) caudal vertebrae lacking filiform haemapophyses. The sister-group relationship between *Anurognathus ammoni* and *Vesperopterylus lamadongensis* is supported by one synapomorphy: char. 271 (2), the complete loss of mid-cervical ribs.

Previous analyses had recovered disparate results. The results of *Wang et al. (2005)*, derived from the matrix of *Kellner (2003)*, indicated a basal position for *Anurognathus ammoni*, as the sister-group of a trichotomy comprising *Batrachognathus volans*, *Jeholopterus ningchengensis* and *Dendrorhynchoides curvidentatus*, which thus comprised the Batrachognathinae according to this topology. The relationship between *Batrachognathus volans*, *Jeholopterus ningchengensis* and *Dendrorhynchoides curvidentatus* was based on the following synapomorphy: a very large humerus, with a humeral/femoral length proportion over 1.40 (*Kellner, 2003*; *Wang et al., 2005*).

This ratio (humeral/femoral length proportion) equals 1.2–1.25 for *Anurognathus ammoni*, 1.43 for *Dendrorhynchoides curvidentatus*, 1.52–1.55 for *Jeholopterus ningchengensis*, and 1.93 for *Batrachognathus volans* (Table 2). As such, it can be seen that the value for *Dendrorhynchoides curvidentatus* and *Jeholopterus* are not that large, not quite close to *Batrachognathus* but actually closer to the one found in *Anurognathus*. Furthermore, all anurognathids subsequently described exhibit such ratios under 1.40: *Vesperopterylus lamadongensis* (1.35) and *Luopterus mutoudengensis* (1.28). Thus, all other anurognathid specimens, irrespective of their ontogenetic stage, exhibit a humeral/femoral length ratio between 1.2 and 1.55 (Table 2), except for the holotypes of *Sinomacrops bondei* (1.77) and *Batrachognathus volans* (1.93). In order to better investigate the informative value of this morphometric character, we categorized it into discrete states by subjecting a comprehensive morphometric dataset for pterosaurs (see Table S1) to a gap-weighting analysis using the software PAST (see "Material and Methods"). As a result, we found the following categories: humerus/femur length ratio up to 0.6 (state 0), over 0.6 and under 1.6 (state 1) and equal to 1.6 or over (state 2). We found state 2 to correspond to a synapomorphy for the clade *Sinomacrops* + *Batrachognathus*, being exclusive for these two taxa among pterosaurs except for "*Huaxiapterus*" *corollatus* (Table S1).

In the analysis by *Wu, Zhou & Andres (2017)*, a polytypic genus *Dendrorhynchoides* (encompassing *D. curvidentatus* and *D. mutoudengensis*) was not recovered as monophyletic. *Dendrorhynchoides curvidentatus* fell at the base of the group, while *Luopterus mutoudengensis* fell as the sister-group of *Batrachognathus volans*. In this analysis (*Wu, Zhou & Andres, 2017*), the clade comprising all other anurognathids to the exclusion of *D. curvidentatus* was supported by one synapomorphy: a fifth pedal digit phalanx 2 straight, instead of curved as in *D. curvidentatus*. This bone is clearly curved in *D. curvidentatus* (see *Ji & Ji, 1998*) and straight in *Luopterus mutoudengensis*, *Anurognathus ammoni* and *Jeholopterus ningchengensis* (*Wang et al., 2002*; *Bennett, 2007*; *Lü & Hone, 2012*), however, it is unknown in *Batrachognathus volans* (see *Riabinin, 1948*), as well as in *Sinomacrops bondei*, and thus is not informative concerning the position of *Batrachognathus*.

More recently, in the analysis of *Longrich, Martill & Andres (2018)*, also derived from *Andres, Clark & Xu (2014)*, the results recovered *Anurognathus ammoni* as the sister-group of *Jeholopterus ningchengensis*, with *Dendrorhynchoides curvidentatus* as the next successive sister-group, and then *Batrachognathus volans* at the base of the group. *Luopterus mutoudengensis* was not included in that analysis. Such topology is compatible with the one presented here, which differs only by the inclusion of *Luopterus*, *Vesperopterylus* and *Sinomacrops*.

## Phylogenetic placement of the Anurognathidae
### Previous works

The interrelationships of anurognathids have been even more obscure than their intrarelationships. Anurognathids have been included in tens of computed phylogenetic

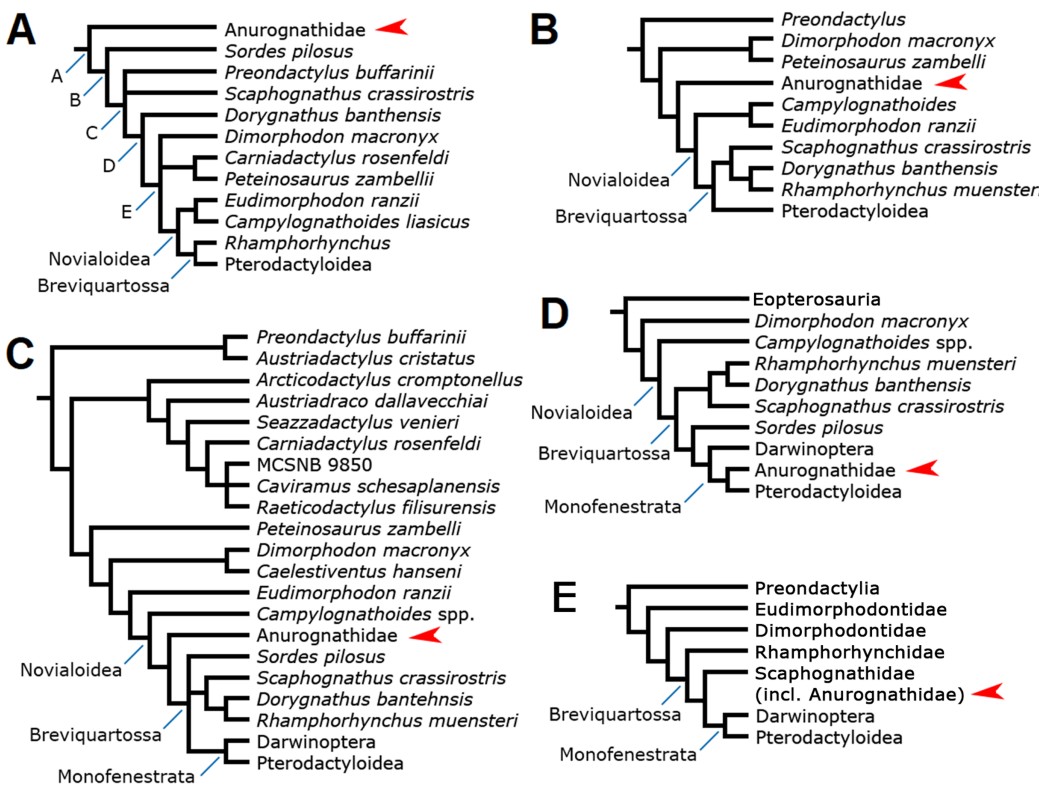

**Figure 11 Previous phylogenetic hypotheses for the position of the Anurognathidae.** Simplified cladograms. (A) From *Kellner (2003)*. (B) From *Unwin (2003)*. (C) From *Dalla Vecchia (2019)*. (D) From *Andres, Clark & Xing (2010)* and *Andres, Clark & Xu (2014)*. (E) From *Vidovic & Martill (2017)*. Red arrows indicate the Anurognathidae.

analyses, although the cladistic hypotheses concerning their placement can be narrowed down to a total of five (Fig. 11).

In the analysis presented by *Kellner (2003)*, the Anurognathidae have been interpreted as the basal-most known pterosaur lineage (Fig. 11A), as the sister-group of a clade containing all other pterosaurs. This result was reproduced by other workers (*Bennett, 2007*; *Lü et al., 2018*). As anurognathids span from the Callovian to the Aptian, this placement would imply in an extensive ghost lineage, as the pterosaur record dates back to the Carnian-Norian (see *Kellner, 2003*). Later versions of this matrix including darwinopterans preserve the same position for the Anurognathidae (e.g. *Wang et al., 2009*). More recent versions of this data set focus solely on eupterodactyloids and do not contain a comprehensive number of non-pterodactyloids (e.g. *Wang et al., 2012*; *Holgado et al., 2019*; *Pêgas, Holgado & Leal, 2019*).

The analyses of *Unwin (1992*, *1995*, *2003)* recovered anurognathids as the sister-group of the clade *Campylognathoides* + Breviquartossa (=Rhamphorhynchidae + Pterodactyloidea), which is equivalent to the Novialoidea sensu *Kellner (2003)* (Fig. 10B). Recent versions of this matrix, covering further non-pterodactyloids (including darwinopterans), preserve the same position for the Anurognathidae (e.g. *Codorniú et al., 2016*). However, it is interesting to observe that *Unwin (2003)* also discussed the possibility

that anurognathids were, in fact, the sister-group of the Pterodactyloidea, although the strict consensus tree ultimately favored their interpretation as the sister-group of the Novialoidea. *Unwin (2003)* noted that anurognathids shared with pterodactyloids a reduction of the cervical ribs and reduction of the caudal series, and stated that a possible close relationship between them was worthy of further investigation. Possible relationships between anurognathids and pterodactyloids had already been discussed also by *Young (1964)*.

The analyses of *Dalla Vecchia (2009*, *2014)* recovered Anurognathidae as the sister-group of the Pterodactyloidea, with *Rhamphorhynchus* as the next successive sister-group. However, these analyses did not include any member of the Darwinoptera (sensu *Andres, Clark & Xu, 2014*). More recently, the subsequent analyses by *Britt et al. (2018)* and *Dalla Vecchia (2019)*, which are more comprehensive (Fig. 11C) and incorporate darwinopterans, have produced a different result, with Anurognathidae being the sister-group of the Breviquartossa (Rhamphorhynchidae + Monofenestrata), and thus within Novialoidea but outside Breviquartossa. A sister-group relationship between the Anurognathidae and the Breviquartossa was also proposed previously by *Viscardi et al. (1999)*.

Under the hypothesis first put forward by *Andres, Clark & Xing (2010)*, the Anurognathidae are monofenestratans and are closer to pterodactyloids than darwinopterans and rhamphorhynchids (Fig. 11D), thus being comprised within the Breviquartossa and the Monofenestrata. This proposition thus echoed the suspicion put forward by *Unwin (2003)* that anurognathids could, perhaps, be closely related to pterodactyloids; as well as the past results from *Dalla Vecchia (2009*, *2014)* that were later modified (*Britt et al., 2018*; *Dalla Vecchia, 2019*).

The most recent hypothesis was put forward by *Vidovic & Martill (2017)*, whose phylogenetic analysis recovered the Anurognathidae as a clade comprised within Scaphognathidae (Fig. 11E). Similar to the proposal of *Dalla Vecchia (2014*, *2019)*, this hypothesis also places anurognathids within breviquartossans and outside the Monofenestrata. However, *Vidovic & Martill (2017)* expressed concerns about this result for anurognathids, noting that "*[t]heir deeply nested placement within Scaphognathidae is likely to be due to a lack of transitional-morphs combined with their paedomorphism*" (*Vidovic & Martill, 2017*, p. 9). They further noted that "*[t]he paedomorphic characters exhibited by anurognathines (e.g. reduced rostrum length, large orbit, deep skull, shorter caudal vertebrae) might be the reason some researchers (e.g.* Kellner, 2003; Wang et al., 2009*) find them as the most basal taxa in Pterosauria*" (*Vidovic & Martill, 2017*, p. 9).

It is worth noticing that anurognathids have also been regarded as possibly related to *Dimorphodon* (*Kuhn, 1967*; *Wellnhofer, 1978*), based mainly on similarities in skull shape (high skull with a convex dorsal margin in lateral view, and a subvertical quadrate). No computed phylogenetic analyses have recovered a close relationship between dimorphodontids and anurognathids, so far.

In summary, among all proposed hypotheses, three of them converge in recognizing a clade that includes Rhamphorhynchidae, Anurognathidae, Darwinoptera and

Pterodactyloidea (*Andres, Clark & Xing, 2010*; *Vidovic & Martill, 2017*; *Dalla Vecchia, 2019*), though disagreeing on the relationships between these subgroups. Two hypotheses (*Andres, Clark & Xing, 2010*; *Vidovic & Martill, 2017*) converge in recovering anurognathids as members of the Breviquartossa. Only the phylogenetic analyses of *Andres, Clark & Xing (2010)* found support for the monofenestratan nature of anurognathids, although, prior to the discovery of darwinopterans, *Unwin (2003)* had already expressed some consideration towards anurognathids being the most closely related group to pterodactyloids.

### Present work

Our dataset combines discrete characters coming from previous contributions (*Kellner, 2003*; *Unwin, 2003*; *Dalla Vecchia, 2009*, *2019*; *Andres, Clark & Xing, 2010*; *Andres, Clark & Xu, 2014*; *Vidovic & Martill, 2017*). According to the present results, anurognathids are basal monofenestratans, and thus are also members of the Novialoidea and of the Breviquartossa. As our results have produced a novel topology, this warrants some discussion.

According to our results, anurognathids exhibit the following synapomorphies of the Novialoidea:

Character 192 (0). Dentition, variation in crown shape along the upper jaw: absent; and char. 193 (0) for the lower jaw (*Unwin, 2003*, char. 19; *Dalla Vecchia, 2019* char. 37 and char. 38 for the lower jaw). Remarks: the secondary loss of heterodonty (which is present in basal pterosaurs) had already been recovered previously as a synapomorphy of the Novialoidea (*Andres, Clark & Xu, 2014*; *Dalla Vecchia, 2014*, *2019*).

Character 340 (1). Postacetabular process of the illium length: shorter than preacetabular process (*Vidovic & Martill, 2017*, char. 212). This feature had already been recovered as a synapomorphy of the Novialoidea (*Vidovic & Martill, 2017*). It can be seen in *Dendrorhynchoides* (*Ji & Ji, 1998*), *Jeholopterus* (*Wang et al., 2002*) and *Anurognathus* (*Bennett, 2007*).

Character 380 (2). Pedal digit V, phalanx 2, length: shorter than preceding phalanx (*Vidovic & Martill, 2017*, char. 195). This feature is primitive for novialoids as seen in *Campylognathoides* (*Padian, 2008b*), *Sordes* (*Unwin & Bakhurina, 1994*), *Scaphognathus* (*Bennett, 2014*), darwinopterans (*Wang et al., 2010*) and pterodactyloids (see *Vidovic & Martill, 2017*). It is present in *Jeholopterus* and *Luopterus* (*Wang et al., 2002*; *Lü & Hone, 2012*), although it is reversed in *Dendrorhynchoides* and *Anurognathus* (*Ji & Ji, 1998*; *Bennett, 2007*).

Anurognathids further share with the Breviquartossa the following synapomorphies:

Character 48 (1). Premaxilla extending to orbit, but no further. This feature had already been recovered as a synapomorphy of the Breviquartossa by *Unwin (2003)*. This feature can be seen in *Anurognathus* (*Bennett, 2007*).

Character 147 (1). Mandible, surangular eminence: absent (*Unwin, 2003*, char. 16). Remarks: the secondary loss of this feature had already been considered a synapomorphy of the Breviquartossa (*Unwin, 2003*). The feature is absent in *Anurognathus ammoni* (*Bennett, 2007*) and cannot be assessed in other species.

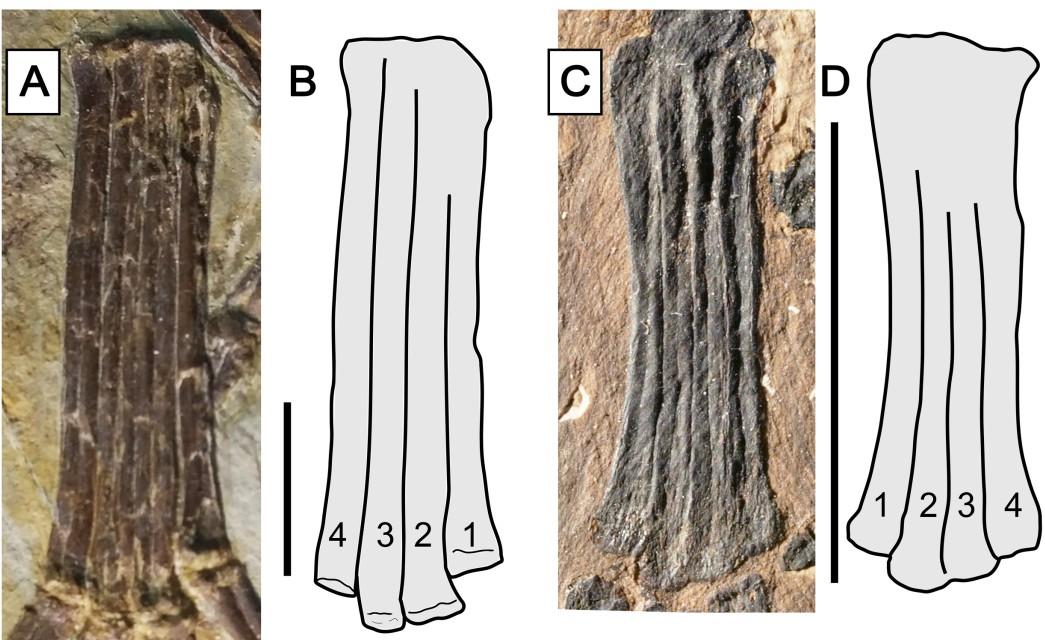

**Figure 12 Metatarsus in anurognathids.** (A) *Vesperopterylus lamadongensis* holotype BMNHC-PH-001311. (B) Schematic drawing. (C) *Jeholopterus ningchengensis* specimen CAGS IG 02-81. (D) Schematic drawing. Numbers refer to metatarsals. Scale bar equals 10 mm.

Character 179 (2). Dentition, distal teeth, spacing relative to successive teeth: more than diameter of teeth (*Andres, Clark & Xu, 2014*). This character had already been recovered as a synapomorphy of the Breviquartossa (anurognathids included) by *Andres, Clark & Xu (2014)*.

Character 284 (1). Humerus, shaft, cross-section: tapered (*Andres, Clark & Xu, 2014*). Remarks: this feature, as opposed to a subcircular cross-section of the humeral shaft, has already been recovered as a synapomorphy of the Breviquartossa, anurognathids included (*Andres, Clark & Xu, 2014*).

Character 368 (1). Fibula, relative length: shorter than tibia (*Dalla Vecchia, 2009* char. 68, modified from *Unwin, 2003* char. 8). Remarks: this feature has already been recovered as a synapomorphy of the Breviquartossa, including anurognathids, by *Dalla Vecchia (2009)*.

Char. 373 (2). Metatarsals, relative length of metatarsal IV: shorter than metatarsals I–III (*Unwin, 2003*, char. 21). This feature has already been recovered as a synapomorphy of the Breviquartossa, and the clade name actually derives from this feature (*Unwin, 2003*). In anurognathids, this feature can be seen in *Vesperopterylus* and *Jeholopterus*, although metatarsal IV is only slightly shorter than metatarsal III (by, approximately, the width of their diaphyses; Fig. 12). The length difference is thus less conspicuous than in *Rhamphorhynchus* or *Scaphognathus* (*Wellnhofer, 1975*, *1978*), but similar to that seen in *Sordes* (*Wellnhofer, 1978*), *Darwinopterus* (*Lü et al., 2009*), *Pterodactylus antiquus* or *Diopecephalus kochi* (*Wellnhofer, 1970*, *1978*). The feature is lost in *Anurognathus*, in which metatarsals I–IV are subequal in length (*Bennett, 2007*).

Character 378 (1). Pedal digit V, phalanx 2, shape: curved (*Kellner, 2003*, char. 74). Remarks: primitively, this phalanx is straight, as seen in non-breviquartossans such as *Campylognathoides* (*Wellnhofer, 1978*; *Padian, 2008b*), *Dimorphodon* (*Padian, 1983*), and Triassic forms (*Dalla Vecchia, 2014*). The phalanx is curved in rhamphorhynchids (*Wellnhofer, 1975*, *1978*; *Lü et al., 2012*; *Hone et al., 2012*), *Dendrorhynchoides* (*Ji & Ji, 1998*) and *Kunpengopterus* (*Wang et al., 2010*; *Cheng et al., 2017*), and changes to "bent, angled" (state 2 of same character) in some taxa such as *Dorygnathus*, *Scaphognathus* and *Darwinopterus* (*Andres, Clark & Xu, 2014*; *Vidovic & Martill, 2017*; *Dalla Vecchia, 2019*), and reverses to "straight" (state 0) in the *Luopterus–Jeholopterus–Anurognathus* clade, in which this phalanx is straight (*Wang et al., 2002*; *Lü & Hone, 2012*; *Bennett, 2007*; *Andres, Clark & Xu, 2014*).

Our analysis has also recovered the Digibrevisauria, coined by *Vidovic & Martill (2017)* for a clade that comprises the Scaphognathidae and the Monofenestrata, to the exclusion of rhamphorhynchids. Anurognathids show the following features that were recovered as synapomorphies of the Digibrevisauria: 236 (1) proximal caudal vertebrae lack distinct lateral processes; 275 (2) humerus deltopectoral crest not as long as the humeral head is wide (seen in *Sinomacrops* and *Batrachognathus*, reversed to state 1 in the clade containing the remaining anurognathids); 313 (1) metacarpal IV lacks a *crista metacarpi*; 375 (1) phalanges of pedal digit IV unequal in length with the distal phalanx larger than all those preceding it, and 376 (1) phalanges 2 and 3 of pedal digit IV are squared or shorter than they are wide (*Vidovic & Martill, 2017*).

Within digibrevisaurians, anurognathids were recovered as basal monofenestratans. The Monofenestrata have been phylogenetically defined by *Andres, Clark & Xu (2014)* as a synapomorphy-based clade, defined by the presence of a confluent nasoantorbital fenestra synapomorphic with the one seen in *Pterodactylus antiquus*. In summary, considering the interpretation put forward by *Andres, Clark & Xing (2010)* that anurognathids possess a nasoantorbital fenestra (corroborated here), this would mean that the clade Anurognathidae + (Darwinoptera + Pterodactyloidea) corresponds to the Monofenestrata. According to our results, thus, anurognathids are basal monofenestratans. The Monofenestrata were recovered based on the following four features:

Character 15 (1): Confluent nasoantorbital fenestra. Remarks: most workers have coded a confluent nasoantorbital fenestra as absent for anurognathids (*Kellner, 2003*; *Unwin, 2003*; *Bennett, 2007*; *Lü et al., 2018*; *Vidovic & Martill, 2017*), except for *Andres, Clark & Xing (2010)*, *Andres, Clark & Xu (2014)* and *Dalla Vecchia (2019)*. Due to the extremely reduced preorbital region and the small absolute size of anurognathids, investigation of their preorbital fenestration is indeed difficult. In most specimens, the situation cannot be confirmed, such as the holotypes of *Jeholopterus ningchengensis*, *Dendrorhynchoides curvidentatus*, *Luopterus mutoudengensis* and *Vesperopterylus lamadongensis*, and also the specimen NJU–57003. The only specimen for which a skull element was tentatively interpreted as an ascending process of the maxilla (and thus a bony bar effectively separating naris and antorbital fenestra, as two distinct openings) is the second specimen of *Anurognathus ammoni* (*Bennett, 2007*). The identification of this process has been reviewed and challenged by *Andres, Clark & Xing (2010)*, who argued that the purported

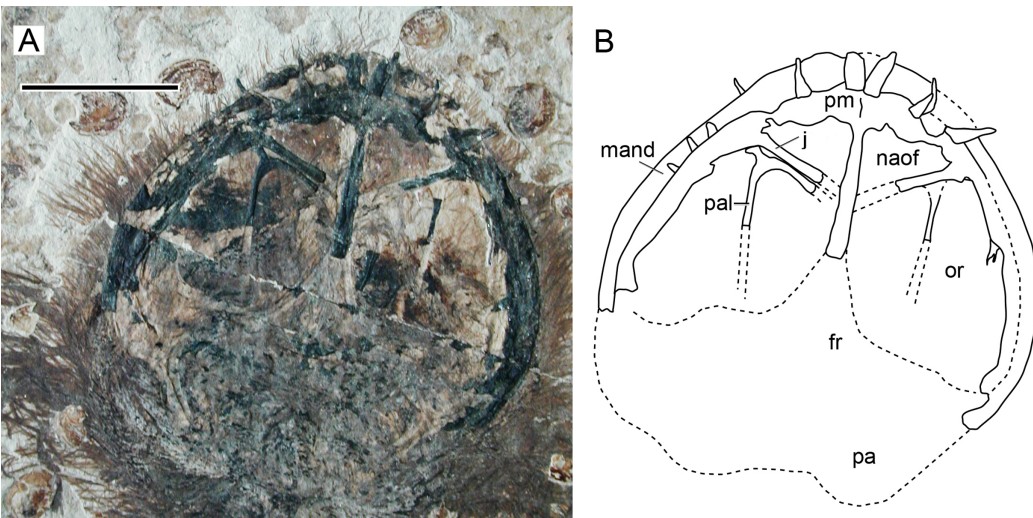

**Figure 13 Nasoantorbital fenestra in *Jeholopterus* CAGS IG 02-81.** (A) Skull exposed in dorsal view, and (B) schematic drawing. Abbreviations: fr, frontal; mand, mandible; max, maxilla; naof, nasoantorbital fenestra; or, orbit; pa, parietal; pal, palatine; pm, premaxilla. Scale bar equals 10 mm.

process could not be unequivocally identified as an ascending maxillary process separating the nares from the antorbital fenestra, as it could only be seen on the right side, was a faint impression, and was displaced, so that even its natural orientation cannot be unambiguously assessed. Based on its rough location and shape, we offer a tentative interpretation for it as a palatal element. *Andres, Clark & Xing (2010)* further noted that there are two previously described anurognathid specimens in which the preorbital region is well preserved and the ascending processes of the maxilla is absent on both sides: the holotype of *Batrachognathus* and CAGS IG 02-81 (see *Riabinin, 1948*; *Ji & Yuan, 2002*; *Andres, Clark & Xing, 2010*; *Yang et al., 2019* and also Fig. 13). In accordance, in the small preorbital region of *Sinomacrops*, only a single opening is present. We thus favor the interpretation of *Andres, Clark & Xing (2010)* that a nasoantorbital fenestra is present in anurognathids (Fig. 13).

Character 95 (1). Jugal, lacrimal process, subvertical. Remarks: this feature has already been recovered as a synapomorphy of a clade containing Monofenestrata + *Sordes* (*Andres, Clark & Xu, 2014*). In the present analysis, we coded this character as "anteriorly inclined" (state 0) for *Sordes* (as in the dataset from *Vidovic & Martill (2017)*), so that the feature is restricted to the Monofenestrata.

Character 216 (1) Atlantoaxis fusion. Remarks: this feature has already been recovered as a synapomorphy of the Monofenestrata, including anurognathids (*Andres, Clark & Xu, 2014*). This feature is present in *Anurognathus* (*Wellnhofer, 1975*; *Bennett, 2007*).

Character 221 (1). Mid-cervical vertebrae, ribs: short. Remarks: as already noticed before (*Unwin, 2003*), the reduction of mid-cervical ribs can be seen in anurognathids and pterodactyloids. Short mid-cervical ribs have been reported for *Jeholopterus* (see *Wang et al., 2002*) and are absent (state 2 of this same character) in *Anurognathus* and

*Vesperopterylus* (see *Bennett, 2007*; *Lü et al., 2018*). The mid-cervical ribs are also short (and quite slender) in the Darwinoptera (*Wang et al., 2009*, *2010*; *Cheng et al., 2017*).

Finally, Darwinoptera + Pterodactyloidea is supported by the following features that are absent in anurognathids: char 1 (1) elongated skull, over four times skull height (*Dalla Vecchia, 2019*, char. 1), char. 112 (1), the craniomandibular joint is located under the orbit (and not posterior to it), char. 230 (0) first dorsal rib larger than others (*Vidovic & Martill, 2017*, char. 236; homoplastic with *Eudimorphodon*), char. 311 (2) pteroid over 2/5 ulnar length (*Dalla Vecchia, 2019* char. 70), char. 317(0) metacarpal IV posterior crest absent (*Vidovic & Martill, 2017* char. 164; present in *Dendrorhynchoides*, see *Ji & Ji, 1998*), char. 366 (1) femur less than twice the length of metacarpal IV (*Kellner, 2003* char. 71; homoplastic with Rhamphorhynchini, *Eudimorphodon*, *Fenghuangopterus* and *Sinomacrops*), 370 (1) splayed metatarsals (*Dalla Vecchia, 2009*, char.70; homoplastic with rhamphorhynchids, *Sordes* and *Scaphognathus*), and char. 375 (3) distal and proximal phalanges of pedal digit IV longer than those between (reversing to state 1, proximal phalanx is the largest, in the Pterodactyloidea).

In summary, these results provide support for the inclusion of the Anurognathidae within the Breviquartossa and, more specifically, within the Monofenestrata (as in *Andres, Clark & Xing (2010)* and *Andres, Clark & Xu (2014)*), though not closer to pterodactyloids than darwinopterans. In this way, these results represent a new hypothesis for the position of the group, being somewhat intermediate between the results of *Andres, Clark & Xing (2010)* and of *Dalla Vecchia (2009*, *2019)*. Still, as well-put by a reviewer (N. Jagieslka), pterosaur phylogeny is a "*fluid, ever-expanding field*", and as noted by *Vidovic & Martill (2017*, p.9*)*, studies of anurognathid phylogeny are hampered by their "*aberrant morphology*". Thus, much work will be needed before the phylogenetic position of anurognathids stabilizes (hopefully with the discovery of "transitional-morphs"), although the present results do lend support for their interpretation as monofenestratans.

## A remark on " *Dimorphodon weintraubi*"

This is a North American Pliensbachian taxon, represented by a partial skeleton still mostly undescribed (*Clark et al., 1998*) and awaiting a detailed description. If "*D. weintraubi*" is taken into consideration, it is recovered as the immediate sister-group of the clade containing all other anurognathids (*Dalla Vecchia, 2009*, *2014*, *2019*, present work). If Anurognathidae is considered as a branch-based clade (sensu *Hone, 2020*; the most inclusive clade containing *Anurognathus* but not *Scaphognathus*, *Dimorphodon* or *Pterodactylus*), then "*D. weintraubi*" would be a basal anurognathid. This relationship is supported in our analysis by two synapomorphies: char. 326 (0) first wing phalanx under 0.35 total wing digit length, and char. 331 (2) wing phalanx 3 shorter than phalanx 1. According to the results by *Britt et al. (2018)* and *Dalla Vecchia (2019)*, they also share a boot-like prepubis. "*D. weintraubi*" further exhibits a conspicuously shortened metatarsal IV (*Clark et al., 1998*), typical of the Breviquartossa.

If this relationship and our new results are correct, then "*D. weintraubi*" pushes the origin of the Monofenestrata back to the Early Jurassic (Pliensbachian). The Early-Middle Jurassic pterosaur record is rather scanty, and the diversity of monofenestratans during

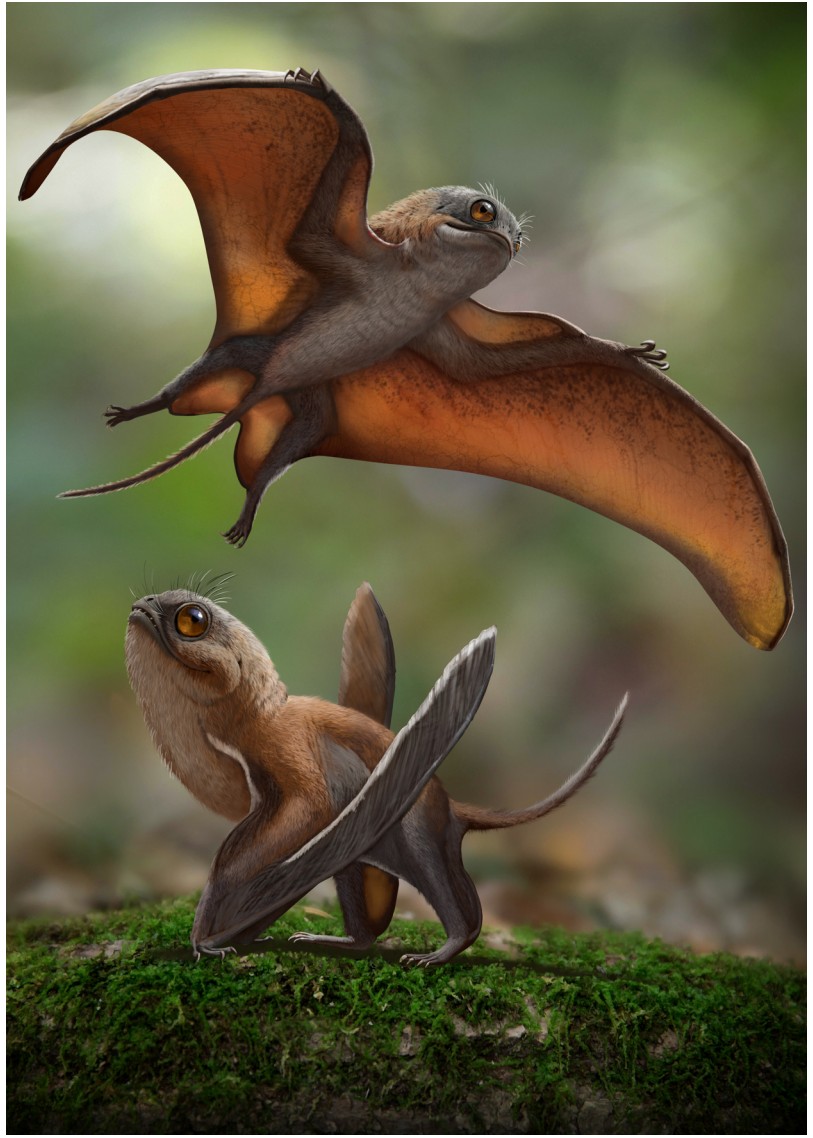

**Figure 14 Life reconstruction of *Sinomacrops bondei*.** Paleoart courtesy of Zhao Chuang, reproduced with permission.                                                   

that time might have been higher than previously thought. Such scenario is not that farfetched, given that the sister-group of the Dibigrevisauria, the Rhamphorhynchidae, dates back to the Toarcian. A detailed redescription and reassessment of "*D. weintraubi*" is of the uttermost importance.

## CONCLUSIONS

JZMP-2107500095 represents a new anurognathid, here named *Sinomacrops bondei* (Fig. 14). It is the second anurognathid from the Tiaojishan Formation, and the first anurognathid specimen to exhibit a skull exposed in lateral view. In our new phylogenetic analysis, it is recovered as the sister-group of *Batrachognathus volans*, with which it comprises the Batrachognathinae. All other taxa were recovered as closer to *Anurognathus*.

The exclusion of *Luopterus mutoudengensis* from the genus *Dendrorhynchoides* is corroborated. *Vesperopterylus lamadongensis* is recovered as the sister-group of *Anurognathus ammoni*, with *Jeholopterus ningchengensis* as their successive sister-group.

Some previous interpretations of anurognathid morphology and systematics have relied on limited available information. With time and new specimens being discovered, new data have been provided and new interpretations were presented. For this reason, each new specimen is crucial for the understanding of the group. The present information available leads us to interpret anurognathids as basal members of the Monofenestrata, as the sister-group of Darwinoptera + Pterodactyloidea.

## INSTITUTIONAL ABBREVIATIONS

**BMNHC**   Beijing Museum of Natural History, Beijing, China
**BYU**   Brigham Young University Museum of Paleontology, Provo, Utah, USA
**IVPP**   Institute of Vertebrate Paleontology and Paleoanthropology, Beijing, China
**JPM**   JZMP, Jinzhou Museum of Paleontology, Jinzhou
**NJU**   Nanjing University, Nanjing, China

## ACKNOWLEDGEMENTS

We thank the Willi Hennig Society for making TNT freely available. RVP thanks Maria E. Leal (Aarhus University) for discussions. RVP and ZX deeply thank Niels Bonde (Zoological Museum, Copenhagen) for his heartful support and for fostering our network. We thank Natalia Jagielska, Steven Vidovic and an anonymous reviewer for their thorough and constructive remarks.

### Funding

This work was supported by the National Natural Science Foundation of China (grant #41688103, #41672019 and #41790452), the National Key R&D Program of China (2019YFC0605403), the China Geological Survey (DD20190397), and the FAPESP (#2019/10231-6). The funders had no role in study design, data collection and analysis, decision to publish, or preparation of the manuscript.

### Grant Disclosures

The following grant information was disclosed by the authors:
National Natural Science Foundation of China: #41688103, #41672019 and #41790452.
National Key R&D Program of China: 2019YFC0605403.
China Geological Survey: DD20190397.
FAPESP: #2019/10231-6.

### Competing Interests

The authors declare that they have no competing interests.

## Author Contributions

- Xuefang Wei conceived and designed the experiments, performed the experiments, analyzed the data, prepared figures and/or tables, authored or reviewed drafts of the paper, and approved the final draft.
- Rodrigo Vargas Pêgas conceived and designed the experiments, performed the experiments, analyzed the data, prepared figures and/or tables, authored or reviewed drafts of the paper, and approved the final draft.
- Caizhi Shen conceived and designed the experiments, performed the experiments, analyzed the data, prepared figures and/or tables, authored or reviewed drafts of the paper, and approved the final draft.
- Yanfang Guo conceived and designed the experiments, performed the experiments, analyzed the data, prepared figures and/or tables, authored or reviewed drafts of the paper, and approved the final draft.
- Waisum Ma conceived and designed the experiments, performed the experiments, analyzed the data, prepared figures and/or tables, authored or reviewed drafts of the paper, and approved the final draft.
- Deyu Sun conceived and designed the experiments, performed the experiments, analyzed the data, prepared figures and/or tables, authored or reviewed drafts of the paper, and approved the final draft.
- Xuanyu Zhou conceived and designed the experiments, performed the experiments, analyzed the data, prepared figures and/or tables, authored or reviewed drafts of the paper, and approved the final draft.

## Data Availability

The new specimen (holotype of a new species, Sinomacrops bondei) is housed in Jinzhou Museum of Paleontology, (Jinzhou, China) under the accession number JPM-2012-001.

The CT scan of JPM-2012-001 is available at MorphoSource: DOI 10.17602/M2/M165765.

Raw measurements are available in Table 1 and in the Supplemental Files.

## New Species Registration

The following information was supplied regarding the registration of a newly described species:

Publication LSID: urn:lsid:zoobank.org:pub:15997DEB-0EF7-40F6-80B0-2C40ED47D43B.

*Sinomacrops* gen. nov. LSID: urn:lsid:zoobank.org:act:C1268C7D-80AA-4854-93E7-0E60220A05BC.

*Sinomacrops bondei* gen. et sp. nov. LSID: urn:lsid:zoobank.org:act:048E9ADE-8C3A-47D4-B074-DCEFA40BDE9A.

## Supplemental Information

Supplemental information for this article can be found online at http://dx.doi.org/10.7717/peerj.11161#supplemental-information.

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
