# Peer review of "Sinomacrops bondei, a new anurognathid pterosaur from the Jurassic of China and comments on the group"

_PeerJ, doi:10.7717/peerj.11161_

## Round 0.1 · original submission · Major Revisions

The reviewers found serious weaknesses in your work. You should give your full attention to their comments.

Please, together with your unmarked revised manuscript, provide a marked-up copy as well as a document explaining how you have addressed each of the points raised by all three reviewers.

·

Basic reporting

The paper is well structured, has natural flow and progression with well-defined easy-to-navigate sections. The discussion is extremely detailed and accurately literature supported throughout. The figures are clear and support statements presented well (with one noted exception). There are some minor revisions however I will note down.
The abstract flows well and covers the aims and results of the paper nicely. Albeit I have some minor comments that might increase readability.

1. On line 29 the specimen is described “first anurognathid skull exposed in lateral view”, while it is the best-preserved skull in lateral view on record; arguably BSPG1922.1.42 or Anurognathus holotype crania is also preserved in lateral view. The preservation of that specimen is suboptimal due to heavy calcite replacement and missing elements. Better rephrase to “best-preserved anurognathid skull exposed in lateral view”, although arguably crania of your JPM-2012-001 looks to be more in ventrolateral view rather than lateral. Line 279 – again, the cranium looks to be exposed in ventrolateral view, albeit due to the heavy deformation this does not necessitate urgent correction.

2. Lines 31-33, “As a result, we obtain a new hypothesis for the placement of the group, somewhat intermediate between other two recent proposals” is a little confusing, no proposals were noted prior in the abstract. Define the proposals: anurognathids being at the base of Pterosauria or sister to pterodactylia (or any other major phylogenetic take you explore in this study)

3. On line 24, you use scansorial and arboreal interchangeably. While these are connected these are not interchangeable, scansoriality is locomotion type while arboreality is habitat occupation. A creature can be scansorial but not arboreal and vice versa. Adaptability to scansoriality is not mentioned later in the paper, but it is on arboreality (line 44) (Bennett 2007, Witton 2013, Lü et al. 2018). I would suggest just retaining arboreal and removing scansorial.

Other minor corrections can increase readability and be more approachable. The references used are comprehensive, recent, and well placed within the text, albeit I do have some minor suggestions noted below.

1. On point 331 “(This alveolar) pattern is unprecedented for anurognathids”, it would be beneficial to describe alveolar spacing in other specimens rather than just reference papers citing these.

2. Placing pterosaurs within any order of Archosauria is extremely arguable, note which paper you used to place it within the archosauromorph clade (there’s also Avemetatarsalia & Ornithodira and such). I would recommend citing Hone & Benton, 2007, or more recent study.

3. For fossil record range, rather than using popular science book (Witton, 2013)(line 1133) maybe reference study mentioning Azdarchid bottleneck or other more recent papers (recommending Csiki-Sava, 2015 for extinction period, or the start of record by Andres, 2014).
Witton book while being a great resource, is a simplified, now somewhat outdated account that should be replaced with more direct studies.
Use of Witton, 2013 also happens on line 42, summarising synapomorphies, I would recommend Hone 2020 (already cited in your paper) as it does a better job for this purpose.

4. You reference in your methodology using TNT dataset by Dalla Vecchia (2019), albeit that paper copies methodology of Britt 2018 (also referenced in your paper), with the only major change being the addition of Seazzadactylus. It would be beneficial for clarity and reproducibility to cite both, as one creates matrix and goes in-depth on methodology rather than just addition of a clade.
5. 643 small formatting issue

The figures are excellent, clearly annotated and help with visualisation and exploration of characters. With absolutely wonderful artwork and clear line arts.

1. I would suggest the addition of a table of measurements of bone elements (with comparative taxa), and, if possible, stratigraphic section (I very strongly advise this for increase in study reproducibility and future research!)
2. .A large aspect of the phylogeny relies on the elongation of the caudal series, in this specimen the caudals are not well preserved, nor well documented on the photograph. Can high resolution close under right light be provided to make this essential phylogenetic aspect more visible?

Experimental design

The study provides a throughout and valid look into anurognathid phylogeny with the addition of characters to a reputable established taxonomic dataset. The autapomorphies mentioned are valid and the specimen will definitively be welcomed as one of best Anurognathids on record. I am looking forward to the complimentary study on soft-tissue preservation. There is no comprehensive analysis of anurognathid phylogeny, and this study does not repeat too many elements of the recently published study by Hone (2020) also covering diversity in the clade. The TNT file is working, replicates results noted in the text and is sufficiently annotated in Mesquite to allow replication. However, I do have some considerations and corrections (and suggestions) to figures, methodology and facts in the text.

1. One must bear in mind the Dalla Vecchia/Britt matrix relies on phylogenies established by Uwin and Kellner in 2003, at the eve of digital phylogenetic studies. Few characters included might be ontogenetic or act as poorly defined character bins (notably character 16 (Lacrimal, shape in lateral view); character 68 (DPC humerus shape) on which this study heavily relies. While Britt/Vecchia is one of better phylogenies in pterosaur literature it comes with its limitations and requires serious reworking, as attempted by Andres or Vidovic (well explored in your study with appropriate mention of the utilisation of continuous characters on the cranium and similar). It will be worth acknowledging these shortcomings in discussion and taken into consideration when assigning new clade names. Pterosaur phylogeny is a fluid, ever-expanding field and pterosaur phylogeny is overdue large reworking.
2. Study relies on features and relative lengths of the caudal series. One of synapomorphies of newly defined clade of Anurognathini is “the loss of filiform processes of the caudal zygapophyses and haemapophyse” (lines 227-233). While this is true to most anurognathids, there are exceptions within the Novialoidea. Notably Bellubrunnus rothgaengeri (Hone, 2012) which “lacks elongate chevrons and zygapophyses”. Bellubrunnus was omitted from your phylogenetic study, its addition could push entire Anurognathid clade lower into the Novialoidea from current Wukongopteridea sister-condition if filiform processes are treated as discrete character. Much like anurognathids the pterosaur also has concave terminal wing phalange (albeit WP4 is not as reduced). Bellubrunnus also retains large orbit relative to cranial length (although this might be ontogenetic feature). I strongly recommend coding this taxa in. One cannot also forget posterior four bones in caudal section of most non-pterodactyloids lack zygapophyses too. It is not an exclusively pterodactyloid and anurognathid trait and that should get mentioned in some capacity. Even, one specimen of Changchengopterus pani (Lu, 2009) is known for having reduced caudal prezygapophyses and postzygapophysis. Lu also suggested the specimen was a “a basal member of rhamphorhynchoids, and more closely related to Dorygnathus than to other rhamphorhychoids”. These were early phylogenetic studies however without notion of mosaic evolution within the clade.

 Hone, D.W., Tischlinger, H., Frey, E. and Röper, M., 2012. A new non-pterodactyloid pterosaur from the Late Jurassic of Southern Germany. PloS one, 7(7), p.e39312.
 Lu, J., 2009. A new non‐pterodactyloid pterosaur from Qinglong County, Hebei Province of China. Acta Geologica Sinica‐English Edition, 83(2), pp.189-199.


3. The study claims using Dalla Vecchia (2019) matrix used in this study mentions not treating any characters as ordered. The original methodology (as in Britt, 2018) “Three multistate characters were treated as ordered (character numbers 63, 75 and 92)”, this might affect final reading. Include ordered and unordered options should be run as a precaution test
4. You note fusing of cranial elements (and state of scapula coracoid), noting relative ontogenetic stage is subjective but might help in future and assignment of growth stages for this clade. After all, it is known ontogeny has a strong effect on cranial, dental, humeral, and public morphology on Novialoidea (Bennett, 1995). It would also be helpful to have relative element lengths provided in a table, with other related taxa as a comparison.
5. Presentation of Wukongopteridea being an exclusively Oxfordian taxon in Figure 7 & line 144 is untrue. The earliest evidence of Wukongopteridea comes from Bathonian Great Oolite group in the UK – this comprises of a segment of rostrum with antorbital fenestra and sagittal crest (NHMUK PV R 464), a cervical vertebra (NHMUK PV R 40126a), along with two metacarpals (MUM STR1244b, NHMUK PV R 28160b)(O’Sullivan, 2015; O’Sullivan, 2018) these elements are too partial to be included in the phylogeny. Nevertheless, this pushes the range of wukongopteridea to Bathonian (instead of Oxfordian). This suggests fully developed bauplan by Middle Jurassic. More basal members should therefore be expected in later periods extending wukongopteriod ghost lineage by few million years.

O'Sullivan, M., 2015. The taxonomic diversity of British Jurassic pterosaurs (Doctoral dissertation, University of Portsmouth).
O'Sullivan, M. and Martill, D., 2018. Pterosauria of the Great Oolite Group (Middle Jurassic, Bathonian) of Oxfordshire and Gloucestershire, England. Acta Palaeontologica Polonica, 63(4), pp.617-644.

Validity of the findings

The conclusions are well stated and reasonable, albeit more comments have to be made on the limits of the fossil record and used taxonomic parameters.

1. The autapomorphies noted on 253 are reasonable. But, to quote Hone (2020): “one could make a credible case for either synonymising many of these putative genera and species, or if such limited traits are considered sufficient to diagnose taxa, elevating current species to genera and naming most of the unnamed specimens as new genera”. With poor preservation of specimens, usually at contorted angles, without knowledge of ontogeny, it is hard to accurately propose discrete species with a degree of accuracy. This might only be improved or refined with time and more specimens of good quality.

The specimen and proposed clade names fall within ICZN requirements, although I do hope more experienced reviewer can double-check this.

Comments on data, and suggestions for some rewording and acknowledgements of result validity – both in text and TNT matrix.

1. Newly introduced character (ch37 on line 173) is possibly ontogenetic and might provide false signals and make subadult specimens sort in a same arbitrary category. Effects can be accentuated by small numbers of other valid characters to counteract the ontogenetic (or taphonomic) effect.
2. Synapomorphy on line 238. Could poor preservation and taphonomic deformation affect shaft morphology? With specimens being so brittle taphonomic alteration to mandible morphology or any long bone dimensions or elements should be considered. Deformation or taphonomic influence have not been mentioned in the study and should when defining a small number of characters on variably preserved specimens.

There is no major unreasonable speculation outside of the record.

Additional comments

The entire paper is well worded, easy to follow and is first of such nature: exploring phylogeny of this interesting pterosaur clade. Along with a very robust, good description of the novel holotype, supported by useful figures. The conclusions are reasonable and sufficiently explored in the text, albeit more discourse about result validity would be beneficial.

Reviewer 2 ·

Basic reporting

This paper has some major issues that need to be attended to. All of these are covered in the attached marked-up document so I will not go into detail here but stick to the main points.

The diagnosis of the new genus is very weak. There are only 2 traits, one of which is clearly incorrect (the tibia is not unusually long as your own data show) and the other is not easy to code or see in most other taxa. As a result the support is very poor. This is then compounded by a complete lack of comparisons to any of the other names anuroganthids and any examination of their diagnoses (which were recently revised in Hone, 2020 which you cite but then ignore with areas like this). As such, despite the quality of the specimen, this is really not supported as a new taxon as described here.

The description is OK but the figures are not as clear as they could be. Various traits that are described are no visible on either the photos of line drawings which makes them hard to determine. If the authors are not even confident to draw them on their figures this is difficult to credit when reading the text.

Secondly, the phylogenetic analysis is not well explained. There needs to be much more information on the support for various nodes in terms of numbers of characters and what those character states are.

As a related point, the revised diagnosis is unnecessary. Naming a new clade supported by only a few traits in a highly unstable clade that changes in almost every analysis is just creating a name that will never survive even a year until another analysis overturns it. The other re-diagnoses of clades (including of the Anuroganthidae itself) are equally problematic for the same reason. This is actively making the taxonomy worse, not better.

The results are over reliant on the foundational analysis. That is not to criticise the choice here, but pterosaur systematics are notorious for varying greatly between different matrices and failing to recognise this is a major issue. These taxonomic revisions are all based on the assumption that this analysis is 'the' answer and really it is only one and will be contradicted by the next published analysis (indeed I know of two in the works by different groups that produce results different to those produced here). In short, these results need to be held in the context of the previous analyses and some awareness of how tenuous these are likely to be. (Take for example the coding of a confluent NAOF which is controversial in anuroganthids, and I am personally very confident is wrong, but this is glossed over her).

The long section discussing previous results of anuroganthid relationships is over long and large chunks can be taken out without losing anything important. As per the point about, some context would be welcome. Some of the criticism of papers from 2003 for example, that have different numbers and types of characters, and differently formulated OTUs fail to account for what this would have meant for coding at the time and the results produced. Pointing to errors in coding is one thing, but this is repeatedly framed as somehow proving that the results of analyses based on there were incorrect when of course that need not necessarily be the case. One would need to recheck and revise all codings and run the analysis again with the same taxa and methods, simply assuming alternate trees would be produced or the published ones not supported is incorrect. In short this section is over long, and is overly critical. This would be far better replaced with a discussion of the traits that do support your analysis and consideration of alternate codings for things like the NAOF than pointing out coding errors in 15 year old papers.

Experimental design

As per the point about, the methods are basically fine, but the reporting of the results is insufficient. The naming and revised diagnoses of the clades are poorly done.

Validity of the findings

Basically OK but the new genus is not well defined, and the 'assumption' that this result is totally correct and used as a platform for some overly harsh criticism of other analyses is misplaced.

Additional comments

The timing of this paper was clearly unfortunate coming out so soon after the Hone 2020 review. However, since you clearly saw this paper since you cited it and used the next taxonomy that included Luopterus, it is bizarre that you otherwise ignore it. A major review of the entire clade including a new diagnosis of the clade, revised characters of every genus and descriptions of every known specimen, descriptions of the anatomy, and more and yet you do not refer to any of this and often ignore or change things without any reference to it. I suggest you read it and cite it appropriately, currently you have clearly not done so.

Annotated reviews are not available for download in order to protect the identity of reviewers who chose to remain anonymous.

·

Basic reporting

Writing
Generally, this is a well written manuscript with few errors in the language.

There are typographical errors on lines: 31, 471, 539, 769, 788, 828
There are missing words or a poor use of language on lines: 103, 470, 497, 635, 764, 944
On lines 167 and 168 the authors confuse the species names belonging to their respective genera. “Dendrorhynchoides mutoudengensis and Luopterus curvidentatus” should be Dendrorhynchoides curvidentatus and Luopterus mutoudengensis.

In the results section, the way the clade Anurognathinae is reported reads like it is a stepwise succession from Dendrorhynchoides up to Anurognathus + Vesperopterylus. The authors should rewrite this section so that it is clear that Luopterus and Dendrorhychoides are in a monophyletic clade which is the sister group to Jeholopterus + (Anurognathus + Vesperopterylus) - if they find the same results once the corrections below are made.

Other problems with the manuscript include:
The list of institutional abbreviations does not list all of the abbreviations used throughout the paper.
There are missing elements from the list of abbreviations in figure 2 i.e. fe and mt
Figure 7 claims to be the strict consensus with a relationship recovered in a compromise tree indicated by a dashed line. This is an inappropriate representation of the tree, which is in fact a 50% majority rule compromise tree, but the relationships recovered in less than 100% of the most parsimonious trees are indicated by a dashed line.
References given in the text are not included in the reference list (e.g. Vidovic and Martill, 2017 - given the volume date 2018 in the submission) or recorded incorrectly (e.g. Jiang et al. 2014).

Literature review
The literature review is sufficient to give a reader new to the field a reasonable understanding of what work has been done on the group before, but it is incomplete in terms of understanding relevant work on pterosaur phylogeny and there are concerning oversights.

The authors claim their submission is the second to study all known anurognathid species in a cladistic analysis, seemingly unaware of Unwin’s early study of basal pterosaurs which included Anurognathus, Batrachognathus, Dendrorhynchoides and Jeholopterus (Unwin, 2003a) prior to Vesperopterylus and Luopterus being described. Other authors followed Unwin, meaning I have been able to identify at least 14 studies that came before this one which included all known anurognathid species at the time (Kellner, 2004; Wang et al., 2005, 2008, 2009; Lü and Ji, 2006; Lü et al., 2006; Dalla Vecchia, 2009; Lü, 2009; Andres, 2010; Andres, Clark and Xing, 2010; Andres and Myers, 2012; Andres, Clark and Xu, 2014). If the authors consider the possible anurognathid Mesadactylus important with respect to studying all anurognathids in a cladistic analysis, Bennett did not suggest the synsacrum may make it an anurognathid until 2007 (Bennett, 2007), meaning there were six studies considering all known anurognathids at the time of publication before this one.

In the discussion, the authors refer to Kellner’s (2003) cladogram as the first cladistic hypothesis of anurognathid interrelationships. However, Kellner’s work was contemporaneous with Unwin’s (2003b), being published in the same pterosaur themed volume. Furthermore, there were earlier studies which used computed cladistic analysis to place the Anurognathidae (Unwin, 1995; Kellner, 1996; Viscardi et al., 1999) and their phylogeny had been considered for decades earlier (Young, 1964; Kuhn, 1967; Wellnhofer, 1975, 1978; Unwin, 1992).

It is notable that the holotype specimen of Anurognathus ammoni Döderlein, 1923 has a cranium preserved in lateral view, while the authors claim that the specimen reported in this submission is the first anurognathid to have a cranium preserved in lateral view. This gives me cause for concern about the rigour of the work done and the authors knowledge of the literature they cite.

The authors should consider making a reference to a study on pterosaur feeding habits which specifically addressed anurognathids insectivory (Ősi, 2011).

Accuracy of reporting
There are several unfortunate issues with the rigour and accuracy of the reporting which let this manuscript down.

In the methodology section, it is reported that the characters are “unweighted” which is inaccurate because one cannot run a maximum parsimony analysis with unweighted characters. I think that the authors mean to say that equal weights were used?

In the results section, 321 is given as the tree length for the 9 most parsimonious trees (MPTs), but when I ran the matrix I recovered trees of 319 steps. If the reporting is accurate to what the authors found, I suspect the limited search function got stuck searching an island of tree morphospace. However, using a more robust search strategy, or testing the results sufficiently would have exposed this problem. I reran the analysis with a more robust new technology search with ratcheting to avoid becoming stuck in local optima and the shortest tree length was still recovered as 319. The tree differs from the one reported; most significantly, the clade containing Dendrorhynchoides and Luopterus is not recovered. In the same section, the ensemble consistency index and ensemble retention index (i.e. reported as CI and RI, not ci and ri) are given as ranges, which is confusing because they should be the same for all 9 MPTs. The minimum possible number of steps for this matrix will always be 168 and as long as the shortest recovered tree length is 319, the CI will always be 0.526.

These inaccuracies in the reporting are related to concerning problems with the experimental design and findings.

Experimental design

This submission studies a specimen of anurognathid pterosaur which is new to science. The specimen exhibits few but significant enough anatomical differences to be considered an entirely new taxon. However, it should be noted that proportions can be affected by allometric growth and the spacing between the mesial teeth is unknown in some other anurognathids. The study also purports to present a critical review of anurognathid phylogenetic relationships already in the literature. In places, this review descends into ‘why we’re right and they’re wrong’ ad hominem argumentation. It is my opinion that the general experimental design (not the execution) is sufficient to phylogenetically place the specimen and to lend a little consensus based support to the derived non-pterodactyloid theory (Andres, Clark and Xing, 2010), but it is by no means a gold standard to benchmark other cladistic analyses against.

The authors have supplemented the Dalla Vecchia (2009) basal pterosaur analysis with characters known to be synapomorphies and apomorphies for anurognathids. Similarities represented as synapomorphies are not the only information that leads to a groups placement in an analysis using maximum parsimony, it's also important ‘what the group is not'. I would feel more convinced of the authors strong argumentation for preferring their ‘new’ phylogenetic hypothesis if they were using the most comprehensive dataset studying the taxa to date, rather than simply incorporating some characters with transformations known to be associated with the group into a preferred analysis. Furthermore, there are numerous categorical characters describing continuous data in this analysis for which the states appear to have been chosen arbitrarily or based on the researchers’ opinions and therefore, unintentionally, their biases. Ideally, these characters should be modified into continuous characters, or at the very least one of the many objective methods for assigning categorical states to continuous data should be employed (e.g. Thiele, 1993; Strait, Moniz and Strait, 1996; Rae, 1998).

With regard to one ‘continuous’ character, the authors say “in our analysis, we modified the character from Kellner (2003), now considering a humeral/femoral length over 1.70”. There was little merit in the original character for the reason given above, but the modification of the character in this case is a questionable practice because it seems to be designed to fit the author’s preconceived notion.

The methods are documented in enough detail to be able to rerun the analysis. However, the result cannot be replicated because the optimal MPTs were not reported. Furthermore, the method which is documented is not particularly rigorous considering the availability of functions in TNT (Goloboff and Catalano, 2016).

It is not easy and in some cases impossible to validate the coding in the data matrix supplied. This is mostly due to missing states in the list (e.g. characters 77, 89, 91), but there are also some states which do not make sense - possibly because states have been mixed between characters (e.g. character 25)?

The characters and states which it was possible to read and validate are not well coded in some cases. There is inconsistency, where the authors are willing to speculate and code features which are not preserved but can be inferred from other taxa, requiring a priori decisions about relationships (e.g. characters 1, 2, 5), whereas what can be reasonably inferred or assessed from the specimens is not necessarily coded (e.g. characters 4, 6, 17, 26, 43, 44, 46, 47).
Some other specific concerns about the coding include:
Ch. 3 - The downward curvature of the jaws should not be coded with any certainty because the observed curvature of the maxilla is likely due to taphonomic processes acting on the specimen. Note, the maxilla curves ‘down’ and the dentary curves ‘up’ while the exact opposite occurs in Anurognathus due to the same crushing effect. This character should be coded as unknown for Sinomacrops.
Ch. 7 - I am struggling to understand how the jaw shape in dorsoventral view in Sinomacrops is similar to Batrachognathus and those two are more similar to Scaphognathus than Scaphognathus is to Sordes. I also struggle to understand how this character is coded for some other specimens given their preservation.
Ch. 8 - “Premaxilla,_caudoventral=posteroventral_' present_ absent” - I am not sure what this refers to because the coding is not consistent with my only reasonable interpretation of this character. Please define it better.
Ch.9 - Some anurognathid specimens preserved in ventral view or in a poor state of preservation are coded as having no premaxillary crest, while the authors code it as unknown for this specimen which clearly demonstrates the state is absent.
Ch. 25 - The palatal elements of Sordes are reduced to very thin bars of bone - these elements are not unknown.
Ch. 30 - The authors went to great lengths to convince the reader that the nares were confluent with the antorbital fenestra but then do not code this for Sinomacrops. Although, I note there are characters that duplicate this one and they are coded.
Ch. 36 - It doesn’t make sense that the relative size of the orbit compared to the antorbital fenestra would be coded the same as Dimorphodon in anurognathids. Indeed, if the authors are following their own assessment this character should be coded with a dash.
Ch. 59 - The lack of serrations on distal teeth can be coded for a couple of anurognathids and monofenestratans. The presence/absence of the ascending process of the maxilla as a reference does not seem particularly important.

Validity of the findings

The authors have kindly supplied the CT data for the specimen described in this submission. However, there is a technical barrier to access because the hosting website is reported as a security risk by web browsers. I would appreciate it if this issue could be resolved or if the data could be deposited in another secure open repository.

I am struggling to confirm the distal expansion of wing phalanx 3 which the authors claim to indicate a fourth wing phalanx. I agree there probably is a fourth wing phalanx, but it remains inconclusive.

When discussing the anurognathids exclusion from Novialoidea and Breviquartossa, the authors neglect to report Unwin (2003b) had tested and discussed the hypothesis that anurognathids were the sister group to Pterodactyloidea and presented his cladogram ‘C’ demonstrating this possible relationship. In the assessment of anurognathids as non-breviquartossan novialoids, the final paragraph is written in such a way that it seems to question the validity of Dalla Vecchia’s (2019) findings and synapomorphies, simply because this study did not find the same result. I think this paragraph could be reworked to more plainly state that these relationships and therefore the synapomorphies were not corroborated by this analysis.

Similarly, there are several misrepresentations of the scaphognathid hypothesis. In this submission, the authors discuss numerous synapomorphies which are not coded in the original analysis. Vidovic and Martill (2017) do not code any anurognathids for the character “Angle of quadrate to the dental plain” because it is impossible to measure, even in Anurognathus. Nor do they code “Inferior temporal fenestra/orbit ratio (maximum length)” for any anurognathid.

Also, the authors make the statement “Tooth crowns, recurved”...”was coded as “?” in Rhamphorhynchus, Dorygnathus, Campylognathoides and Dimorphodon” which is incorrect. Vidovic and Martill (2017) studied numerous specimens of these species, detailed in their supplementary material, and coded Rhamphorhynchus, Dorygnathus, Campylognathoides zitteli and C. liasicus, and Dimorphodon ‘0&1’. The authors also argue with correct coding in this analysis. They claim the quadrate position under the orbit was miscoded by Vidovic and Martill (2017) in wukongopterids, saying “all wukongopterids” orbits are anterior to the quadrate. However, the articular end of the quadrate is typically located under the portion of the orbit approximating its centre and often approaching the lacrimal process of the jugal in wukongopterids and basal pterodactyloids. Finally, the authors say “tooth crown curvature displacement”...”is not present in anurognathids, as can be seen from the well-preserved teeth of Anurognathus, Batrachognathus, Dendrorhynchoides and Jeholopterus”. Vidovic and Martill (2017) also coded this state as absent in Anurognathus, but coded it as present in the other 3 species studied. Batrachognathus clearly exhibits strongly curved teeth and Dendrorhynchoides curvidentatus is even named for the curvature of its teeth.

Similar to Unwin (2003b) testing different hypotheses of anurognathid relationships and discussing them, Vidovic and Martill (2017) noted they were unconvinced of their findings which may be affected by a lack of transitional forms and paedomorphism. This warrants some discussion in these sections.

I am unconvinced by the sixth condition which includes anurognathids in Breviquartossa “metatarsal IV shorter than metatarsals I-III”. Because of the preservation, it is difficult to be sure of when the metatarsals III and IV terminate without looking down the microscope, but if metatarsal IV is shorter than the others, it is by much less than the width of its diaphysis. Looking at many other breviquartossans, the fourth metatarsal is significantly shorter than its diaphysis’ width compared to the other metatarsals.

Additional comments

This submission is interesting, describing a new specimen which provides valuable insights into anurognathid palaeobiology. I do not necessarily agree with your assessment of parts of the specimen, in some cases I concede this is down to personal judgement and you are entitled to your opinions. However, in parts the cladistic analysis is coded incorrectly or more confidently than it can be. Furthermore, the methodological approach is insufficient to critically appraise the work of others. The quality of the cladistic analysis, its execution and reporting are left wanting and some of the criticisms made of other studies are unwarranted or not clearly stated as your own opinions. My recommendation is to focus on the specimen and not to get bogged down in trying to fix the phylogeny of the anurognathids.

You must make some changes to your data matrix to ensure the OTUs are coded faithfully and removing subjective alterations to the characters. In the sections above, I have picked out some specific concerns, but I have not comprehensively reviewed the matrix. When you run your analysis, please ensure that you use a robust search strategy.
If you still wish to pursue comprehensively reviewing the phylogeny, please include information about all the hypotheses presented and discussed by Unwin (2017) and Vidovic and Martill (2017) - who were clearly not entirely convinced of their own results but reproduced them faithfully. Also, a much more comprehensive understanding of what has come before is required, but I probably would not have noticed the absence of some studies if the importance of this contribution had not been so overstated in parts.

---

## Round 0.2 · Minor Revisions

Your manuscript is much improved, but the reviewers still raised some concerns.

Please, together with your unmarked revised manuscript, provide a marked-up copy as well as a document explaining how you have addressed each of the points raised by the reviewers.

Reviewer 2 ·

Basic reporting

This is much improved compared to the first version and the major issues have been addressed in the text or response letter. There are still some odd phrases or things that are not clear and I have marked these up and suggested alternatives in the marked up document.

Experimental design

Fundamentally fine. However, I notice now (which I think I missed in the original submission) that all characters are run as unordered even when there are some traits that should be ordered. There is a comment on this and suggested reference in the marked-up document but in short a character that has three states that are e.g. square shape, round shape and triangular shape should indeed be unordered as any transformation from one to the other has no obvious polarity. However, something with states long, medium and short, cannot go from long to short without having 'passed through' medium, this should therefore be ordered. This error is commonly replicated across phylogenetic analyses and should not be perpetuated here.

Given the low support values and constant shifting clade placement in pterosaur phylogenies I would not be surprised if correcting this did change your phylogeny quite significantly and so this really needs to be looked at.

Validity of the findings

Aside from the point about about the stability of the phylogeny this is fine.

Annotated reviews are not available for download in order to protect the identity of reviewers who chose to remain anonymous.

·

Basic reporting

The manuscript is generally in good order with clear English and article structure. The literature cited is sufficient, and provides good context.

There is one example where the authors could unintentionally misrepresent Unwin 2003 to readers without sufficient knowledge of pterosaur discoveries (line 627).

With reference to "Dimorphodon" weintraubi on line 384, I believe the convention of putting a genus in quotation marks indicates that the author is questioning the validity of the genus, whereas, in this case the authors are questioning the specific combination of "Dimorphodon weintraubi".

There appears to be a problem with the reporting of the number of most parsimonious trees recovered from the cladistic analysis. The number currently reported might be historic and needs updating to reflect the trees recovered by analysis of the data presented in the supplementary material. The associated figure appears accurate to my findings.

I am also concerned that there is still a claim (line 787/8) that this new specimen is the first with a skull in lateral view, which is not accurate. I wonder if this is a remnant from a previous draft? I think it is remarkable that this is only the second specimen ever described in lateral view and it is valuable to science as a result.

The abbreviations in the figures need checking. There appear to be some missing.

I have made notes and comments on the PDF of the submission which could help.

Experimental design

The experimental design and associated comparisson with other similar analyses has been revised. The overall quality, readability and accuracy of the manuscript is significantly improved as a result of the well defined research question and relevant results.

I was able to replicate the analysis with ease, but the authors could consider including their search commands in the TNT file for better reproducibility.

Validity of the findings

I personally disagree with the authors' interpretation of the nasoantorbital fenestra, but this is only an opinion informed by the images. I think it is clear that the authors recognise better material is required before a final assessment can be made. I think with the exception of the mistake regarding the occurrence of anurognathids preserved in lateral view, the conclusions are reasonable and based on the findings.

Additional comments

Overall, I think this is a much more coherent paper. There is very little that is objectively wrong with the paper itself and the specimen is a vital contribution to our science.

I notice that one of the figures (13) is stylistically slightly different and I just want to sense check that this is original to this submission? If not, it will need an appropriate citation. If it is original, I don't think the stylisation is so different that it would be necessary to change anything.

I have made notes on the PDF. I have approached this review as succinctly as possible, so I do encourage you to ensure my few comments on the PDF are addressed.

---

## Round 0.3 · accepted · Accept

I confirm that your MS has been accepted for publication.